# Role of grain-level chemo-mechanics in composite cathode degradation of solid-state lithium batteries

Chuanlai Liu [1] ✉, Franz Roters [1] & Dierk Raabe [1] ✉

Solid-state Li-ion batteries, based on Ni-rich oxide cathodes and Li-metal anodes, can theoretically reach a high specific energy of 393 Wh kg$^{-1}$ and hold promise for electrochemical storage. However, Li intercalation-induced dimensional changes can lead to crystal defect formation in these cathodes, and contact mechanics problems between cathode and solid electrolyte. Understanding the interplay between cathode microstructure, operating conditions, micromechanics of battery materials, and capacity decay remains a challenge. Here, we present a microstructure-sensitive chemo-mechanical model to study the impact of grain-level chemo-mechanics on the degradation of composite cathodes. We reveal that crystalline anisotropy, state-of-charge-dependent Li diffusion rates, and lattice dimension changes drive dislocation formation in cathodes and contact loss at the cathode/electrolyte interface. These dislocations induce large lattice strain and trigger oxygen loss and structural degradation preferentially near the surface area of cathode particles. Moreover, contact loss is caused by the micromechanics resulting from the crystalline anisotropy of cathodes and the mechanical properties of solid electrolytes, not just operating conditions. These findings highlight the significance of grain-level cathode microstructures in causing cracking, formation of crystal defects, and chemo-mechanical degradation of solid-state batteries.

Ni-rich layered oxide positive electrodes (cathodes) can yield a substantial increase in energy density of solid-state Li-ion batteries, but they suffer from contact loss, and irreversible layered-to-spinel or disordered rock-salt-like phase transition[1-8]. These structural degradation mechanisms have been mainly attributed to oxygen loss and out-of-plane transition metal (TM) migration arising from cathode/electrolyte interfacial side reactions[3,9-12]. Cathode particle surface modification or coating techniques could partially alleviate interfacial degradation[13-15], however, the effectiveness of these strategies has proven insufficient to mitigate structural degradation in the bulk of cathode materials[2,4,5,16,17]. This dilemma raises the suspicion that the electrochemical instability of the cathode/electrolyte interface may not be the main cause behind the progressive capacity degradation observed over cycling. One of the critical aspects of solid-state batteries is the stress response of their complex microstructure to large volume changes (strains) driven by Li intercalation.

For the layered oxide cathode, Li-ions intercalate into the host structure, creating a gradient in the lattice parameter and a concurrent non-uniform elastic strain and large volume change (7.8% volume change for LiNi$_{0.8}$Mn$_{0.1}$Co$_{0.1}$O$_2$ (NMC811)[8,18,19]). The high-stress buildup resulting from Li transport and crystalline anisotropy inevitably leads to contact loss and the formation of crystal defects in cathodes, in particular dislocations and stacking faults[4,6,20-27], as sketched in Fig. 1. The appearance of crystal defects in ion-insertion materials can generate heterogeneous nanoscale lattice strain and modify the local bonding environment, for example, the number of covalent bonding

[1]Max Planck Institute for Sustainable Materials, Max-Planck-Str. 1, Düsseldorf 40237, Germany. ✉e-mail: c.liu@mpie.de; d.raabe@mpie.de

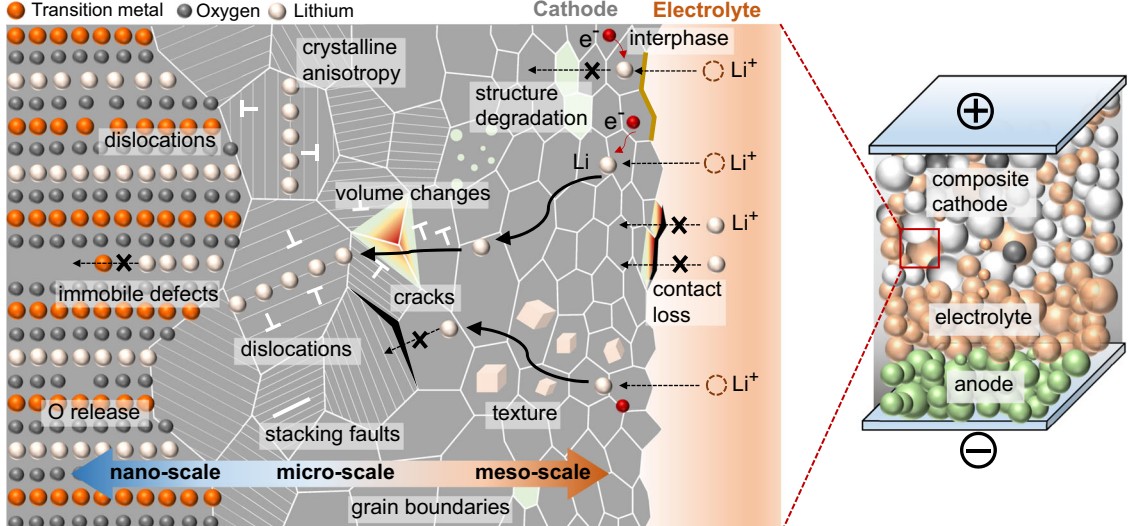

**Fig. 1 | Schematic view of electrochemical reaction, Li intercalation, structural degradation, and lattice defect types in composite cathodes of solid-state battery cells: point defects (zero-dimensional (0D) defects), dislocations (one-dimensional (1D) defects) and interfaces (two-dimensional (2D) defects).** Key to solving the challenges of the cathode/electrolyte interface, crystal defect formation, and structural degradation in cathodes is a clear understanding of the chemo-mechanics of the composite cathodes across battery-relevant length scales. Non-uniform volume changes resulting from Li intercalation and the crystalline aniso-tropy of primary cathode particles lead to high stress, contact loss, and crystal defect accumulation, in particular dislocations and stacking faults. The presence of crystal defects can markedly reduce the energy barriers to remove lattice oxygen, which triggers Li/TM ion mixing and structural degradation. The large composi-tional strains in cathodes result in contact mechanics problems between cathodes and solid electrolytes.

partners for oxygen, as revealed by coherent X-ray diffraction and atomic-scale characterisation[4–6,20–22,26]. The metal-oxygen decoordination owing to defects leads to significant changes in the local electronic structure, and can even change the relative ordering of electronic states[4,22,28,29]. Although the exact nature of the defect-induced bonding environment change is debated and may vary among different mate-rials, such as Ni-rich and Li-rich layered oxides, the large lattice strain associated with crystal defects can markedly reduce the energy bar-riers to remove lattice oxygen and trigger Li/TM ion mixing[4,22,28–31]. Therefore, besides the electrochemical instability of the cathode/ electrolyte interface, the continuous accumulation of crystal defects in cathode particles due to Li (de)intercalation over cycling acts as the primary driving force for aggravating oxygen loss and altering the structural stability.

The solid-state batteries must be able to operate within a wide range of charging and discharging times, ranging from ultra-fast pulses to slow discharging over the course of multiple days. The various compositional strains imposed by these different (dis)charging rates can significantly affect the evolution of grain-level stress and strain responses in composite cathodes. Key to solving the challenges of the cathode/electrolyte interface and crystal defect formation in cathodes is a clear understanding of the chemo-mechanics of the composite cathodes across battery-relevant length scales and strain rates. High-resolution transmission electron microscopy (HRTEM) experiments and atomic-scale simulations reveal that dislocations resulting from heterogeneous Li intercalation can trigger irreversible migration of TM ions into octahedral sites in the Li layer and subsequent structural degradation[4–6,20–22,26,29]. Chemo-mechanical simulations using cohesive zone models and phase-field damage models have advanced the understanding of the microscopic mechanical fracture behaviour in electrode materials[32–44]. However, the assessment of how micro-structure and (dis)charging protocols affect the formation of crystal defects and defect-induced structural degradation at the grain level in composite cathodes remains unexplored. This gap in knowledge is primarily due to the limitations and efforts associated with the appli-cation of advanced and standardised analytical techniques in probing cathodes and their environments across various length scales[4,5].

In this work, we have therefore developed a meso-scale chemo-mechanical constitutive model by integrating the interplay between electrochemical reaction, anisotropic Li-ion intercalation, cathode microstructure, grain-level micromechanics and formation of crystal defects resulting from lattice dimension changes[4,5,28,29]. We apply it to systematically investigate the impact of cathode microstructure, mechanical properties of solid electrolyte, and operating conditions on the evolution of stress and strain responses, generation of dis-locations, as well as the associated oxygen loss and structural degra-dation mechanisms. Our work provides insights into the impact of grain-level chemo-mechanics on the degradation of Ni-rich composite cathodes, aiming at providing a quantitative simulation methodology for mitigating capacity loss of solid-state batteries.

## Results
### Theoretical framework and model setup
The workflow of the electrochemical reaction-diffusion model and dislocation-based micromechanical model, informed by nanoscale experimental and theoretical findings[4,5,12,16,28,29], is shown in Fig. 2. A thermodynamically consistent chemo-mechanical model is derived within the finite strain framework, which is suitable for addressing the large volume change induced by Li intercalation in Ni-rich NMC cathodes[8,18,19]. The model incorporates anisotropic and concentration-dependent material properties, providing an accurate representation of the composite cathode behaviour. An in-house large-scale parallel finite element solver, using the PETSc numerical library[45], was devel-oped to discretize the model and efficiently solve the coupled gov-erning equations[46,47]. The model is implemented in the freeware simulation package DAMASK[48]. The section Methods provides a detailed description of model formulation, numerical implementation, and model parametrization. The procedure comprises three essen-tial steps:

(1) A three-dimensional representative volume element is employed to describe the composite cathode microstructure, as shown in Fig. 2c and Supplementary Fig. S1. NMC secondary particles synthesised by the co-precipitation method typically consist of many randomly oriented primary particles[49,50]. However, modifying the

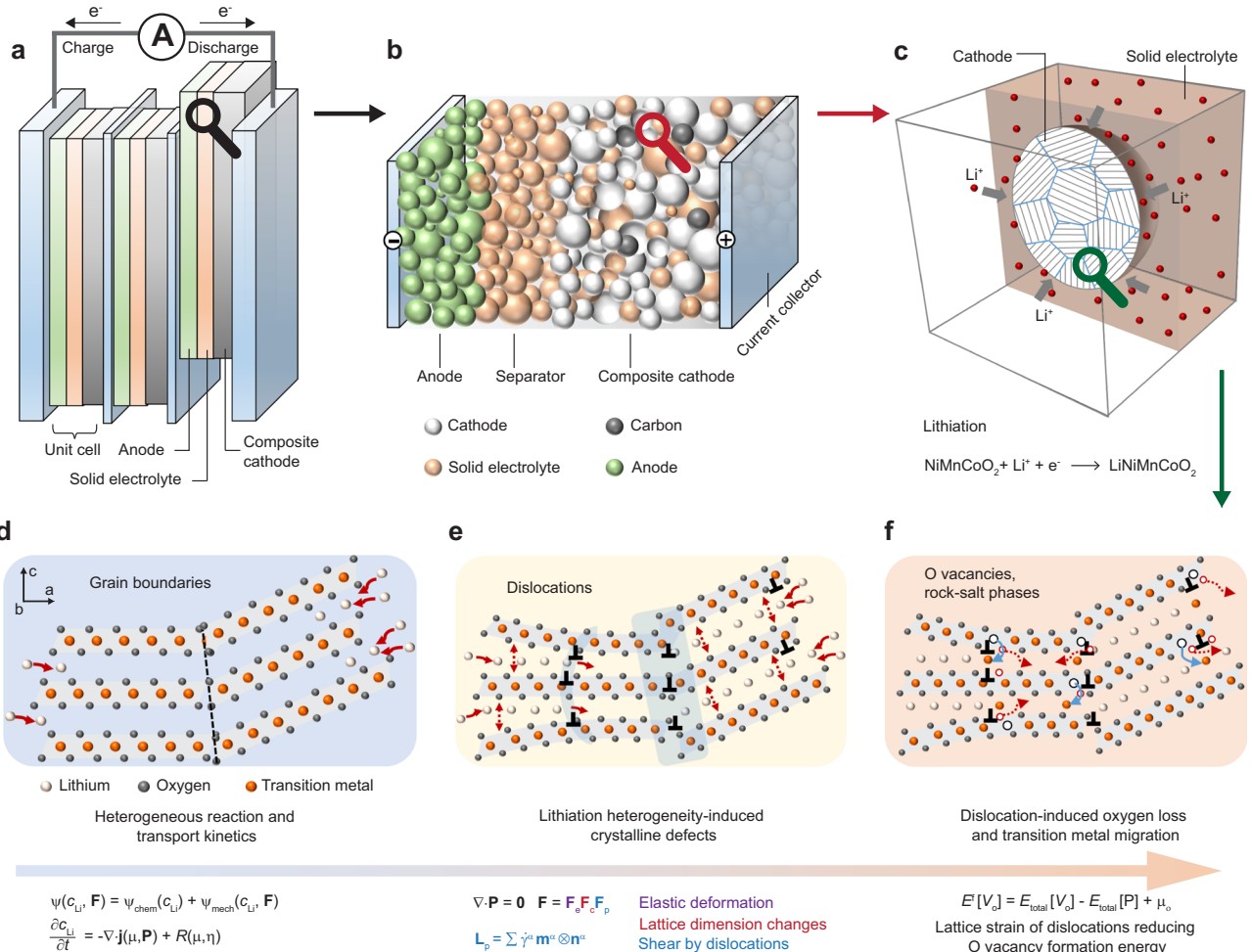

**Fig. 2 | Workflow of the reaction-diffusion-micromechanics and dislocation-induced structural degradation model for composite cathode materials, informed by nanoscale experimental and theoretical findings. a** Concept of a bipolar-stacked Li solid-state battery cell. **b** Schematic of a unit cell featuring composite cathode materials. **c** Representative volume element describing the microstructure of composite cathodes. **d** Application of a Cahn-Hilliard diffusion-reaction model to describe the electro-chemical reactions and Li (de)intercalation in the *a-b* plane. **e** Generation of various crystal defects, such as basal dislocations, arising from the anisotropic lattice dimension changes due to Li (de)intercalation. The coupling of Li composition and dislocation generation is described by a micromechanical constitutive law. **f** The presence of dislocations resulting from Li mass transport facilitates oxygen removal and Li/TM ion mixing.

surface energies through a boron doping strategy can induce the directional growth of primary particles, resulting in NMC secondary particles with radially aligned primary particles[51,52]. In this study, an isolated Ni-rich NMC811 polycrystal particle, consisting of 200 randomly oriented primary particles, is embedded in the uniform solid electrolyte. The polycrystalline cathode particle maintains electrical neutrality as it is assumed to be connected to the current collector via the carbon binder of the composite cathode.

(2) A grain-level chemo-mechanical model is developed to describe the electrochemical reaction, Li intercalation, lattice dimension changes, and dislocation formation. The Cahn-Hilliard reaction-diffusion equation[46,53,54] is employed to describe the electrochemical reaction at the cathode/solid-electrolyte interface, and Li intercalation within the cathode. Li-ion transport inside the solid electrolyte is not considered explicitly since we only simulate (dis)charge processes at constant currents. The composite cathode is working under the galvanostatic discharge or charge condition, *i.e.* a constant Li flux into or out of the NMC secondary particle. A Li occupancy fraction of 0.1 or 0.9 in the NMC cathodes is taken as the stress-free state, respectively. For each primary particle within the secondary particle, the layered structure of the oxide cathode permits Li diffusion exclusively within the basal crystallographic plane, with diffusivity depending on the

state-of-charge (Fig. 3c). To accommodate the anisotropic lattice dimension changes resulting from Li (de)intercalation, misfit dislocations in cathodes are usually generated (Fig. 2d, e), which is described by the crystal plasticity mechanical model[47,55]. Only isotropic elastic deformation is allowed for the solid electrolyte. While the present study uses the maximum principal stress distribution to analyse contact mechanics problems, the current model does not explicitly account for mechanical fracture.

(3) The impact of lattice strain resulting from dislocations on oxygen deficiency in the NMC cathode can be assessed via calculating the formation energy of oxygen vacancies under an applied mechanical strain[4,28,29] (Fig. 2f). Atomic-scale calculations reveal that the formation energy of oxygen vacancies in layered oxide cathodes is significantly reduced when the applied tensile strain approaches 10%[4,22,28,29]. In this study, material domains with a plastic shear exceeding 12% in cathode particles after discharge are categorised as the oxygen-deficient phase. This threshold is determined based on the atomic-scale calculations[4,22,28,29] and by fitting the predicted distribution of oxygen-deficient phase in the secondary particle to experimental characterisation[5]. The formation of such an oxygen-deficient phase in cathode particles will impede the Li-ion intercalation pathways within the cathodes for subsequent cycling. We

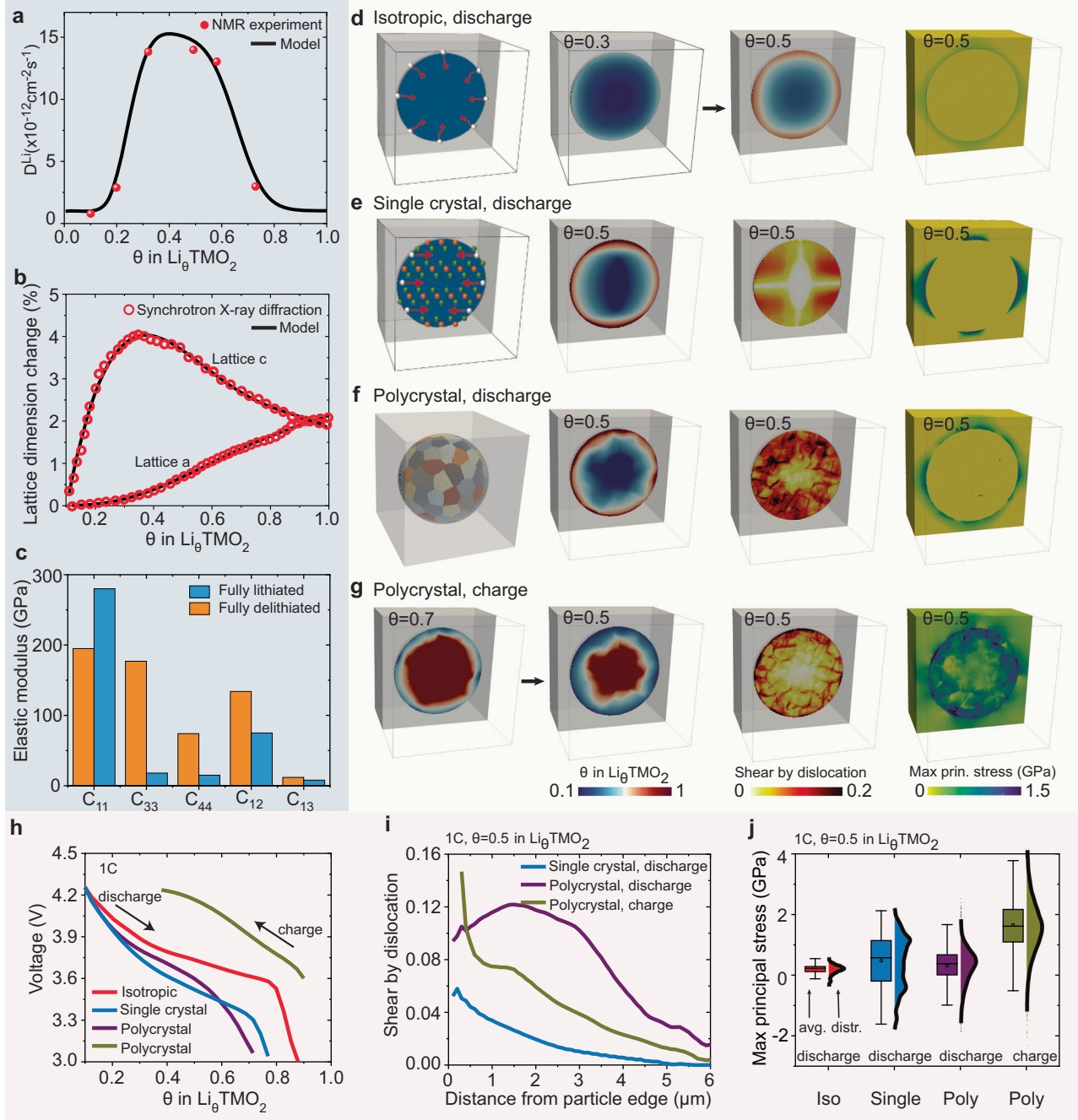

**Fig. 3 | Effect of anisotropic and concentration-dependent electrochemical and mechanical properties of NMC cathodes in Li-ion dynamics, dislocation formation, interface mechanics, and electrochemical performance. a** Li concentration-dependent diffusivity on the basal plane[18]. **b** Anisotropic and Li concentration-dependent lattice dimension changes[18,57]. **c** Elastic stiffness parameters at fully lithiated and delithiated states[58–60]. **d**–**g** Distribution of Li concentration, dislocation-induced shear, and maximum principle stress at 1 C, within a cathode particle with isotropic and constant material properties (**d**), within a single crystal particle (**e**), a polycrystal particle (**f**) under discharge, and a polycrystal particle under charge (**g**), all with anisotropic and concentration-dependent material properties. **h** Voltage-capacity profiles under discharge and charge. **i** The average dislocation-induced shear deformation as a function of distance from the particle edge. **j** Statistical variability in the maximum principle stress distribution at the cathode/solid-electrolyte interface.

assume that the cathode particle is fully delithiated and lithiated when the average Li occupancy in the cathode is 0.1 and 0.99, respectively. The practical absolute discharge capacity of the NMC811 cathode is 203 mAhg⁻¹ [56]. The normalised capacity is defined as the discharge capacity at the cut-off voltage of 3 V normalised to the practical absolute capacity. The total normalized capacity loss at a specific current consists of two components: thermodynamic capacity loss due to the irreversible loss of active materials and kinetically induced capacity loss, which arises from

non-uniform Li distribution within the particles, characterized by Li-rich peripheries and Li-poor cores.

## Role of anisotropic and concentration-dependent electrochemical and mechanical properties

We now investigate the role of anisotropic and concentration-dependent Li diffusivity, elastic stiffness, and lattice dimension changes in Li-ion dynamics, dislocation formation, mechanical failure, and capacity loss in composite cathode materials. A Ni-rich NMC811

particle with a diameter of 12 μm is embedded in the $Li_{6.6}La_3Ta_{0.4}Zr_{1.6}O_{12}$ solid electrolyte. Three types of simulations are performed: (i) a cathode particle with isotropic and constant material properties, (ii) a single crystal cathode particle, and (iii) a polycrystal cathode particle with anisotropic and concentration-dependent material properties. As demonstrated by solid-state nuclear magnetic resonance characterisation, the Li diffusion coefficient drops sharply, over two orders of magnitude, as the Li content exceeds 80%[18,57] (Fig. 3a). The atomic lattice dimension change during (dis)charge is measured by operando synchrotron X-ray diffraction (XRD) experiments[8,18,19], as shown in Fig. 3b. The $a$ and $b$ lattice parameters increase upon lithiation by maximum 2%. The $c$ lattice parameter exhibits a nonmonotonic behaviour, rapidly increasing at the initial stage of discharge by up to 4%, and then gradually collapsing to 1.95% as the Li site fraction exceeds 0.37. These dimensional changes are attributed to the change of Ni oxidation states, and the modification of the interlayer spacing between the $O^{2-}$ planes[8,18,19]. Nanoindentation mechanical experiments and first-principles calculations indicate that delithiation results in a reduction of the elastic modulus of NMC cathodes (Fig. 3c)[58,59]. To the authors' knowledge, the five independent stiffness parameters for NMC811 were only measured or predicted at fully lithiated and delithiated states[58–60]. The elastic stiffness at a fully lithiated state was used here.

Figure 3d–f shows that anisotropic diffusion and state-of-charge-dependent diffusivity result in secondary particles with Li-rich peripheries and Li-poor cores upon lithiation. This heterogeneous Li distribution is insufficient to enable a high Li-ion flux uniformly throughout the particle, thereby leading to a significant increase in overpotential and a consequent sharp reduction in cell voltage during galvanostatic discharge. Therefore, as shown in Fig. 3h, the half cell rapidly approaches the cutoff voltage of 3V, with the inner core of the particle remaining in a Li-deficient state, resulting in a substantial capacity loss. Figure 3h shows that anisotropic and concentration-dependent diffusion results in a normalised capacity loss of 0.12 for single crystal cathodes and 0.18 for polycrystal cathodes at a discharge rate of 1 C, comparing to the isotropic case ($n$C signifies to a full discharge to the practical capacity within $1/n$ hours). Operando optical microscopy observations also confirm the persistent presence of Li heterogeneities within the cathode particle across a wide range of discharge rates[57,61].

Figure 3f, g shows that the large anisotropic lattice dimension change of Ni-rich layered cathodes driven by Li intercalation and the cathode microstructure heterogeneity result in substantial differences in dislocation activity and accumulation between primary particles. This indicates that even primary particles of identical size and orientation will exhibit a degree of dislocation activity that highly depends on their location within the agglomerate. Additionally, the sharp drop in Li diffusivity towards higher Li content conditions leads to a pronounced concentration gradient across the secondary particle. This Li heterogeneity can generate a significant difference in lattice dimensions and distortions between the Li-rich and Li-poor domains. Therefore, as shown in Fig. 3i, basal dislocations accumulate prominently near the exterior of the secondary particle, under both discharge and charge conditions. The formation of crystal defects, such as dislocations, will result in high-stress build-up and very large local lattice strains. This effect, in turn, modifies the local bonding environment for oxygen, ultimately promoting oxygen deficiency[4,5,20–22,26]. The accumulation of these basal dislocations thus facilitates the structural degradation from the layered structure to the spinel-like phase within the agglomerate's periphery[4,5]. This structural degradation carries consequences beyond mere active material loss; it hinders the efficient Li transport into or out of the agglomerate's core. Consequently, this exacerbates the kinetically-induced capacity loss, compounding the adverse effects on the composite cathode's performance.

Figure 3d–g shows that anisotropic elastic stiffness and lattice dimension changes of NMC play a pivotal role in contact mechanics at the cathode/solid-electrolyte interface and grain boundaries among primary particles. Figure 3j shows the statistical variability in the maximum principle stress distribution on the cathode/solid-electrolyte interface (driving force for contact loss), before and after considering the effect of crystalline anisotropy. Figure 3e, f, j suggests that while the cathode particle exhibits volume expansion under discharge, most regions within the cathode/solid-electrolyte interface experience substantial tensile stress due to the anisotropic chemical expansion of primary particles, for both single crystal and polycrystal cathodes. Under the charge condition, Fig. 3g, j shows that the anisotropic deformation of primary particles results in high stresses both at the cathode/solid-electrolyte interface and grain boundaries within the polycrystal cathode particle. This observation indicates that the potential mechanical failure of the cathode/solid-electrolyte interface and primary-particle fragmentation at grain boundaries is driven by the combination of anisotropic lattice dimension changes upon (de) lithiation and microstructure heterogeneity.

## Role of microstructure in rate performance and defect heterogeneity

Insights about the spatial dynamics of Li intercalation and the heterogeneous distribution of crystal defect leverage an improved understanding of the rate performance and electrochemical and mechanical degradation mechanisms of solid-state batteries. Here, we investigate the role of secondary particle size and discharge rate on state-of-charge heterogeneities and dislocation activity in single-crystal and polycrystalline NMC cathodes. Figure 4a shows the predicted and experimental voltage-capacity profiles of the polycrystal cathode during galvanostatic discharge tests[62], where the discharge rate is gradually increased from 0.1 C to 2 C. Furthermore, as shown in Fig. 4b and Supplementary Figs. S4 and 5, for both single crystal and polycrystalline cathodes, the capacity-rate trade-off can be improved by decreasing the secondary particle size of the cathodes. The good agreement between simulations and experiments confirms the effectiveness of the developed physics-based chemo-mechanical model.

Figure 4d suggests that both single crystal and polycrystalline cathodes exhibit similar spatial dynamics of lithiation across a broad range of discharge rates, from 0.25 C to 5 C. However, there are significant differences in dislocation activity between single-crystalline and polycrystalline cathodes. At a low discharge rate of 0.25 C, we observe a high level of dislocation activity in the polycrystalline particle, while the dislocation activity is relatively low in the single crystal particle. Transitioning a high discharge rate of 5 C, dislocations accumulate at the edge of both polycrystalline and single crystal particles; however, the polycrystalline particle exhibits a much broader defect-rich region. Chemo-mechanical phase-field dislocation modelling and operando synchrotron X-ray diffraction experiments demonstrate that the dimensions and geometries of cathode particles remarkably impact the formation of misfit dislocations in phase-transforming cathodes[63–65]. Energy-based stability analysis of misfit dislocations reveals that the minimum critical size for dislocation-free $LiFePO_4$ particles is ~47 nm; below this size, particles are unlikely to host a misfit dislocation at the phase boundary[64]. Synchrotron X-ray diffraction experiments indicate that large misfit strains can be effectively circumvented in electrodes comprising $V_2O_5$ nanospheres with diameters of 49 nm[65]. In the current study, the primary particles in polycrystalline cathodes range from 300 nm to 1 μm in diameter, while single crystal particles range from 2 μm to 12 μm. Thus, the size of the cathode particles is above the minimum critical size of dislocation-free particles[63–65]. Consequently, in this study, plastic shear in the cathodes is mainly driven by anisotropic and heterogeneous compositional strains. The current results suggest that dislocations primarily form as a result of the Li inhomogeneity-induced strain gradient within single

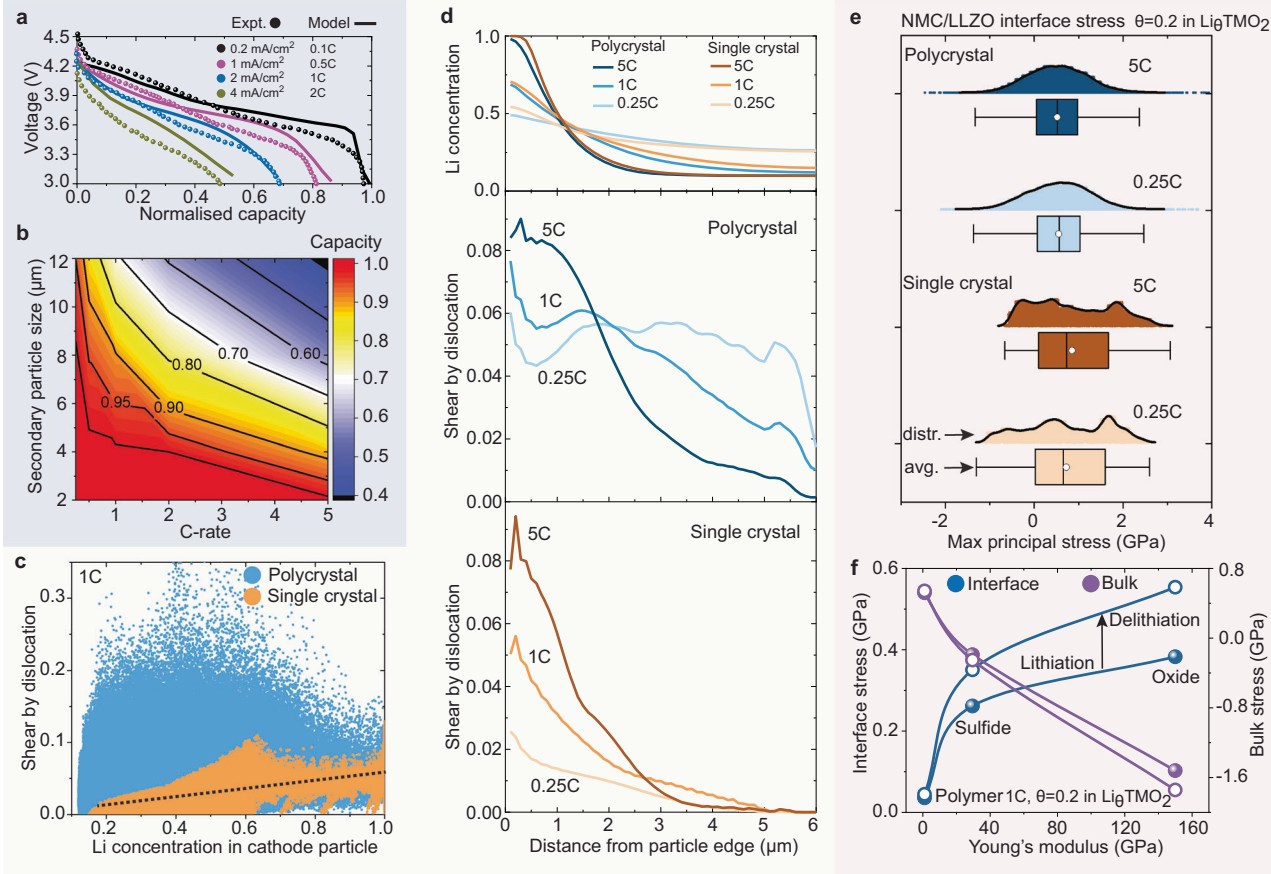

**Fig. 4 | Effect of cathode microstructure and operating conditions on dislocation heterogeneity and mechanical stability. a** Voltage-capacity profiles at various discharge rates for the polycrystalline cathode with a diameter of 12 μm[62]. **b** Effect of the secondary particle size and discharge rate on the capacity of polycrystalline composite cathodes. **c** Correlation between Li concentration and dislocation-induced plastic shear in cathodes. **d** The average Li concentration and plastic shear within the particle as a function of distance from the particle edge. **e** Maximum principle stress distribution at the cathode/solid-electrolyte interface. **f** Effect of Young's modulus of solid electrolytes and electrochemical cycling on the stress response at the cathode/solid-electrolyte interface and within the cathode particles.

crystal particles under high current density conditions. For polycrystalline cathodes, both operating conditions and the random arrangement of primary particles in the secondary particle play a critical role in dislocation heterogeneity. Furthermore, as depicted in Fig. 4c, a positive correlation between the Li concentration and dislocation activity is evident for the single crystal particle, whereas a relatively high dislocation activity is observed in the polycrystalline particle, irrespective of Li content. The particle size-dependent effect on the stability of misfit dislocations should be incorporated into the developed chemo-mechanical model, when the dimension of the electrode particles approaches the critical size[63–65].

Figure 4e and Supplementary Figs. S6, 7 show the distribution of the maximum principle stress at the cathode/solid-electrolyte interface, for both single crystal and polycrystalline particles exposed to various discharge rates. A high tensile stress persists at the interface, irrespective of whether a single crystal or polycrystalline cathode is considered. Moreover, the comparative analysis under different discharge rates in Fig. 4e reveals that the reduction of the discharge rate is not a solution for alleviating this persistent tensile stress at interfaces. However, Fig. 4f shows that reducing Young's modulus of the solid electrolytes instead, through the utilisation of polymer-based or sulfide solid electrolytes, effectively alleviates the high tensile stress at the interface and enhances the overall mechanical stability. Moreover, Fig. 4f shows the effect of electrochemical cycling on the evolution of interfacial stress in composite cathodes. The NMC cathode was initially discharged at 1 C to a cut-off voltage of 3 V and then immediately charged at 1 C. Figure 4f shows that the interface between cathodes and electrolytes underwent significantly higher tensile stress upon charge than discharge. In composite cathodes with oxide electrolytes, the average interfacial stress increased from 378 MPa upon discharge to 580 MPa upon charge at the same state of lithiation.

## Dislocation-induced structural degradation and capacity loss

The presence of crystal defects, such as dislocations and stacking faults, not only induces large lattice strains but also dramatically modifies the local oxygen environment, which manifests itself through inserting extra lattice planes or perturbing the sequence of the oxygen layers[4,5,20–22,26]. Density functional theory calculations reveal that the formation energy of the oxygen vacancy can be markedly reduced from 1.06 eV to 0.24 eV by applying a 10% tensile strain to the layered oxide cathode[4]. Moreover, stacking faults and dislocations can provide an alternative route to form different disordered structures by offering greater freedom to the displacement of TM ions into the alkali metal layers[28,29]. Dislocation-induced irreversible oxygen release and structural degradation are schematically shown in Fig. 5a. This argument is further substantiated by a comprehensive range of characterisations spanning from the atomic to micro-length scale, including HRTEM, Bragg coherent X-ray diffraction imaging (BCDI), and transmission-based X-ray absorption spectromicroscopy and ptychography experiments[4,5,22,28,66]. HRTEM characterisation of a layered oxide cathode after charging shows that pronounced lattice displacements can trigger oxygen loss and TM migration, subsequently leading to a phase

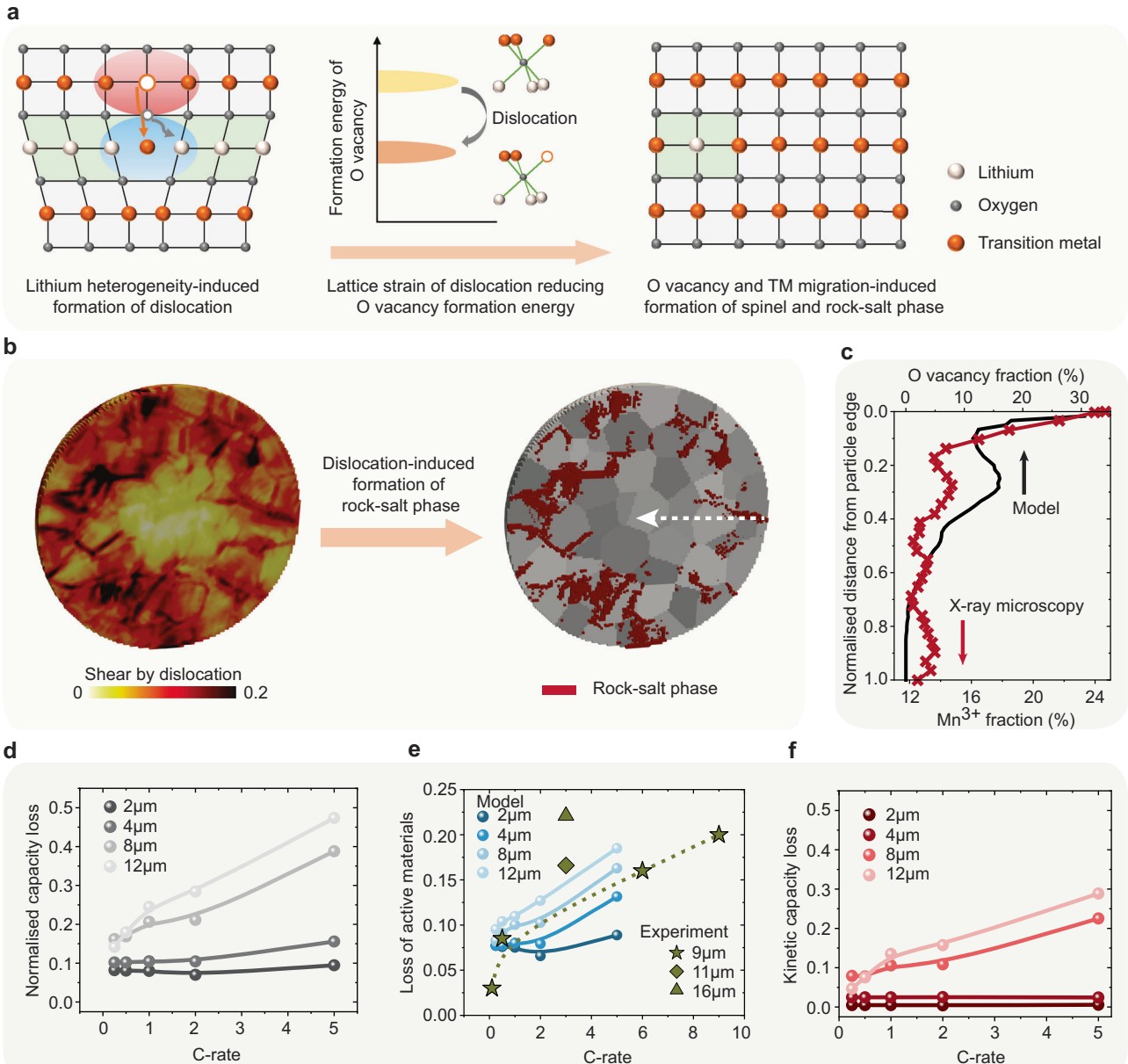

**Fig. 5 | Dislocation-induced structural degradation and capacity loss.**
**a** Schematic of dislocation-induced irreversible oxygen release and structural degradation. Dislocations induce large lattice strain can dramatically modify the local oxygen environment, which can markedly impact the structural stability of the layered phase and trigger oxygen loss and TM migration. **b** Distribution of dislocation-induced plastic shear and oxygen-deficient phase. **c** Comparison of the predicted and measured oxygen deficiency[5]. **d** Total normalised capacity loss as a function of the particle size and discharge rate. **e** Effect of particle size and discharge rate on the fraction of oxygen-deficient phase, compared with experimental characterisation results[67–71]. **f** Kinetically induced capacity loss arising from the impediment of Li-ion intercalation pathways.

transition from the layered structure to the spinel phase[4]. In situ, BCDI measurements illustrate that tensile strain starts accumulating preferentially near the particle surface region and gradually expands into the interior of the particle[4].

Figure 5b shows the predicted distribution of dislocation-induced plastic shear (driven by Li intercalation) in the entire particle after discharge. The inhomogeneous Li concentration distribution and the accumulation of crystal defects significantly affect the structural stability of Ni-rich and Li-rich cathodes, which may ultimately trigger the bulk decomposition of these layered phases. The oxidation state maps obtained through X-ray spectromicroscopy and ptychography reveal that oxygen deficiency persists within the bulk of secondary particles, rather than being limited to the near-surface (a few nanometres) region of the particle[5]. Additionally, these quantitative results show

that the arrangement of primary particles within secondary particles results in notable heterogeneity in the extent of oxygen loss among the primary particles. This observed heterogeneity in oxygen deficiency aligns with the predicted inhomogeneous distribution of plastic shear-induced oxygen loss within the secondary particle, as shown in Fig. 5b. Despite the spatial variation in oxygen deficiency, both experiments[5] and predictions depicted in Fig. 5c consistently indicate that, on average, primary particles located near the exterior of the agglomerate are more susceptible to oxygen loss compared to those in the interior.

Figure 5e shows the effect of the secondary particle size and discharge rate on the fraction of the oxygen-deficient phase within the secondary particle. The results demonstrate that significant bulk structural degradation occurs in secondary particles larger than ~8 μm when subjected to discharge rates exceeding 1 C, leading to a loss of

over 10% of active materials. This emphasises the critical impact of both cathode microstructure and operating conditions on the overall quantity and distribution of the oxygen-deficient phase, a factor that can significantly affect the electrochemical performance of composite cathodes. Figure 5d shows the predicted capacity loss of composite cathodes after accounting for the irreversible oxygen-deficient phase transition in cathodes, as a function of secondary particle size and discharge rate. A noticeable difference in behaviour is observed: particles exceeding 8 μm in diameter exhibit a significant normalised capacity loss, ranging from 0.2 to 0.4, as the discharge rate increases from 1 C to 5 C, while this effect is less pronounced for smaller particles. This phase transition-induced capacity loss is attributed to a combination of factors, including the loss of active materials (thermodynamic effect) and the impediment of Li-ion intercalation pathways within the cathodes (kinetic effect). Furthermore, Fig. 5f shows that secondary particles exceeding 8 μm in diameter and subjected to discharge rates greater than 1 C experience a kinetically induced capacity loss of over 0.1, which is primarily attributed to the accumulation of crystal defects and associated structural degradation at the particle's periphery.

In addition, Fig. 5e includes the effect of cathode particle size and cycling conditions on the loss of active cathode material from experimental characterisation[67–71]. It is noteworthy that the experimental analysis of active cathode material loss generally involves the formation of rock-salt phase and isolated cathode materials induced by intergranular fracture[67–71]. Since this study does not explicitly consider crack formation, only qualitative comparisons are made between predictions and experimental characterisations. Figure 5e shows that the total loss of active cathode material increases with a higher charging rate according to experimental characterisation[67–70], which aligns with the predicted results. For example, increasing the charging rate from 0.5 C to 6 C leads to an increase in the loss of active material from 8.5% to 16%[67–70]. Furthermore, experimental characterisation reveals that the capacity fading of cells using small cathode particles (average diameter of 11 μm) can be reduced from 22.1% to 16.6%, compared to those with large cathode particles (16 μm)[71]. This capacity fading mechanism is generally attributed to the loss of Li inventory, plating of Li, and loss of active material from both electrodes.

## Preliminary insights and perspectives on model development

The current model simplifies or neglects a few other factors that should be addressed to provide a full description of chemo-mechanics in solid-state batteries in the future, for example, explicit description of interfacial and intergranular crack formation[32–35,38–40,43,44,72], concentration-dependent elastic modulus of NMC cathodes[58–60], plastic deformation in solid electrolytes[73–75], role of discrete dislocations and grain boundaries in Li diffusion and chemo-mechanics of NMC cathodes[33,35,76,77], and generation of the representative electrode particle microstructure from microscopy data[43,78].

**Crack formation in composite cathodes.** One of the crucial aspects of composite cathodes in solid-state batteries is the stress response of their microstructure to lattice dimension changes induced by Li transport. As shown in Fig. 3, high tensile stresses build up along grain boundaries in cathodes and at the interface between cathodes and solid electrolytes upon electrochemical cycling. This high stress can lead to contact mechanics problems. Loss of interfacial contact within the cell obstructs Li transport in cathodes and charge-transfer reactions at the cathode/electrolyte interface, which subsequently results in resistance increase, capacity loss, and rate performance deterioration in batteries. While this study effectively captures the heterogeneous stress response and crystalline defect formation induced by compositional strains, it does not account for crack formation. Including mechanical fracture in the model would be beneficial to achieve a comprehensive understanding of the mechanical behaviour

in solid-state batteries. Particle fracture and interfacial contact loss in electrode materials can be integrated into the chemo-mechanical models using cohesive zone models[32–35], spring analogy approaches[38,39], continuum-damage models[40], and phase-field damage models[43,44,72].

**Li-concentration-dependent elastic modulus.** Nanoindentation mechanical experiments and first-principles calculations reveal that the elastic modulus of NMC cathodes is highly dependent on the lithiation state[58,59], as shown in Figs. 3c and 6a. Full delithiation of NMC cathodes significantly reduces their elastic mechanical properties[58,59]. The elastic modulus at fully lithiated and delithiated states thus represents the upper and lower bound for NMC cathodes. Figure 6a, b shows the effect of the elastic modulus of NMC cathodes on the formation of dislocations and stress distribution at the interface between cathodes and electrolytes, at a discharge rate of 1 C. The reduction in the elastic modulus of NMC cathodes leads to a decrease of the average dislocation shear from 0.09 to around 0.03 within the cathodes. However, the maximum principal stress distribution at the interface between cathodes and solid electrolytes is minimally impacted by the decrease in the elastic modulus of NMC. Figure 6b shows that high tensile stress persists at the interface, regardless of whether the elastic modulus corresponds to the fully lithiated or delithiated state.

**Elasto-plastic deformation in solid electrolytes.** The assessment of the chemo-mechanical behaviour of composite cathodes in this study assumes that the solid electrolyte behaves as an elastic solid, which is valid for oxide electrolytes[74,75]. However, sulfide electrolytes are expected to undergo plastic deformation in response to the volume change of electrodes due to their low yield strength[73–75]. For example, indentation measurements indicate that the yield strength for amorphous and crystalline sulfide electrolytes ranges from 200 MPa to 450 MPa, while the yield strength for oxide-based electrolytes is significantly higher, ranging from 2 GPa to 3 GPa[73–75]. Figure 6c–f shows the effect of the plastic deformation in oxide and sulfide electrolytes on the chemo-mechanical behaviour of composite cathodes, at a discharge rate of 1 C. Here, solid electrolytes are modelled as isotropic elasto-plastic materials using the isotropic $J_2$ plasticity model[48], with a yield strength of 2 GPa for oxide electrolytes, 450 MPa for crystalline sulfide electrolytes, and 200 MPa for amorphous sulfide electrolytes[73–75]. Figure 6c shows that oxide-based electrolytes exhibit minimal plastic deformation upon lithiation of composite cathodes. The impact of plastic deformation in oxide electrolytes on the formation of dislocations in cathodes and interfacial stress distribution is negligible (Fig. 6d). In contrast, Fig. 6e, f shows that while the plastic deformation in sulfide electrolytes does not obviously impact dislocation formation in NMC cathodes, it can effectively reduce the tensile stress concentration at the interface between cathodes and solid electrolytes.

**Dislocation-mediated Li diffusion.** Bragg coherent diffraction imaging characterisation[21,22] and chemo-mechanical phase-field dislocation modelling[64,79,80] indicate that discrete dislocations can result in large lattice mismatch and non-uniform stress fields near dislocations in electrode materials. These stress-strain fields around dislocations can potentially alter the overall Li diffusion behaviour and Li concentration distribution[64,79,80]. For example, chemo-mechanical phase-field simulations have demonstrated that there is strong Li enrichment and depletion in the tensile and compressive stress fields around the dislocation, respectively, in $LiMn_2O_4$ cathodes[79]. Additionally, pipe diffusion can be initiated on the tensile stress side of edge dislocations[79]. Moreover, recent advancements in strain engineering have employed dislocations to tune ionic transport properties in Li-ion conducting argyrodites $Li_6PS_5Br$[81]. These findings suggest that inducing internal strain and dislocations in the structure of argyrodites

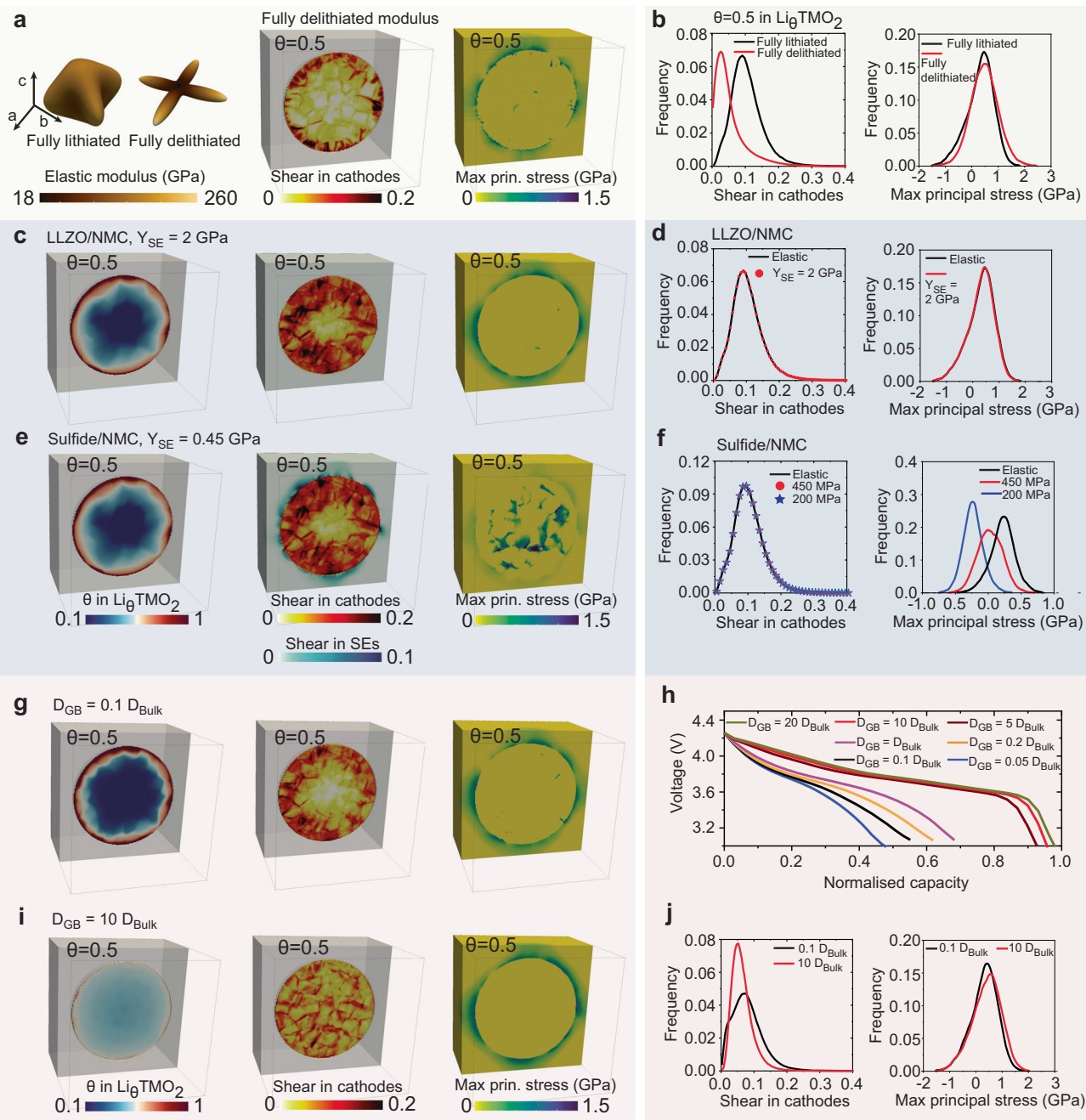

**Fig. 6 | Effect of material properties on the chemo-mechanical behaviour of composite cathodes. a** Elastic modulus of NMC cathodes[58–60]; distribution of plastic shear and maximum principal stress, using the elastic modulus at the fully delithiated state. **b** Frequency plots showing the effect of elastic modulus on the shear in cathodes and interfacial maximum principal stress. **c, e** Distribution of Li concentration, plastic shear, and maximum principal stress, considering the plastic deformation in oxide (**c**) and sulfide (**e**) electrolytes. **d, f** Effect of plastic deformation in oxide (**d**) and sulfide (**f**) electrolytes on the shear in cathodes and interfacial maximum principal stress. **g, i** Distribution of Li concentration, plastic shear, and maximum principal stress, with a tenfold decrease (**g**) and increase (**i**) in Li diffusivity along grain boundaries. **h** Voltage-capacity profiles. **j** Effect of Li transport kinetics along grain boundaries on the shear in cathodes and interfacial maximum principal stress.

through applying uniaxial and hydrostatic pressures can promote Li-ion disorder, and lead to higher Li-ion conductivity[81]. Although the current model in this study does not explicitly describe the effect of self-stress around dislocations on Li-ion transport behaviour in electrodes and solid electrolytes, this effect can be incorporated into the chemo-mechanical model by reformulating the diffusivity tensor as a function of plastic shear.

**Grain boundaries in polycrystalline cathodes.** Grain boundaries, with their distinct physicochemical properties compared to the bulk,

noticeably affect Li transport, mechanical failure, and structural degradation in polycrystalline NMC cathodes[14,20,76,82,83]. For example, the low diffusion barriers of TM ions along grain boundaries can promote TM ion dissolution and accelerate formation of rock-salt phases[14,20,83]. Additionally, the atomistic structures of various grain boundaries, such as local structural distortion and charge redistribution, can significantly influence Li transport kinetics[76,82]. Conductive atomic force microscopy characterisation shows that grain boundaries can generally act as fast pathways for Li transport in $LiCoO_2$[82]. First-principles calculations reveal that the coherent $\Sigma2$ grain boundaries

enhance Li migration both along and across the grain boundary plane, whereas the asymmetric Σ3 grain boundaries substantially impede Li transport in NMC cathodes[76]. The role of grain boundaries in Li transport kinetics can be incorporated into the chemo-mechanical model by considering the chemical potential or concentration jumps across grain boundaries[33,35,77].

In this study, grain boundaries are represented by two layers of elements located between adjacent grains, as shown in Supplementary Fig. S1. Li diffusivity along grain boundaries is assumed to range from 0.05 to 20 times that in the bulk. As shown in Fig. 6g, i, fast Li transport along grain boundaries can significantly enhance the homogeneity of Li distribution within the polycrystalline cathode particle, and effectively decrease the overpotential on the surface of cathodes. Figure 6h shows the voltage-capacity profiles at a discharge rate of 1 C, with various Li transport kinetics along grain boundaries. A fivefold increase in Li diffusivity along grain boundaries results in a 20% increase in capacity, while a corresponding fivefold decrease in Li diffusivity leads to an 8% capacity fade. This indicates that, besides the mechanisms intrinsic to the crystal and electronic structure of NMC cathodes, tailoring and optimising grain boundary properties in polycrystalline NMC cathodes can significantly enhance their rate capacity. For example, solid-state electrolyte infused along the grain boundaries in Ni-rich NMC particles acts as fast channel for Li transport, which realises an increase in capacity retention from 79% to 91.6% after 200 cycles[84]. Figure 6g, i, j shows that high tensile stresses exist at the interfaces between cathodes and solid electrolytes regardless of Li transport kinetics along grain boundaries, which is largely determined by anisotropic compositional strains and mechanical properties of composite cathodes.

**Constructing microstructure from experimental data.** Given the anisotropic electrochemical and mechanical properties of NMC cathodes, the present study, along with chemo-mechanical fracture simulations[43] and experimental characterisation studies[51,52], demonstrate that regulating the morphology and crystallographic orientation of primary particles can effectively mitigate crystal defect formation, stress concentration, and microcracking in Ni-rich NMC cathodes. Such regulation can significantly improve the rate and cycling performance of composite cathodes. For example, modifying the surface energies through a boron doping strategy can induce the directional growth of primary particles, resulting in NMC secondary particles with radially aligned primary particles[51,52]. This unique crystallographic texture allows diffusion channels to penetrate directly from the surface to the centre, significantly improving the Li-ion diffusion coefficient. Moreover, this radial alignment can effectively alleviate the volume-change-induced stress and intergranular fracture in cathodes[43,51,52]. This representative three-dimensional microstructure for chemo-mechanical modelling can be generated based on multimodal microscopy characterisations[78,85,86]. Statistical representations of particle microstructures can be derived from X-ray nano-computed tomography data[85] and sub-particle grain representations can be derived from electron backscatter diffraction data[86].

## Discussion

The pursuit of mitigating capacity loss in Ni-rich oxide cathodes has been a focal point of research, targeting challenges including oxygen loss, structural degradation, and mechanical failure within composite electrodes[1–3,6,7,9,12,16]. The transition from a layered to rock-salt phase in NMC cathodes is an inevitable result of cationic mixing and oxygen loss[87]. Among Ni, Co, and Mn ions, Ni ions have the strongest tendency to mix with Li ions, primarily due to the similarity in ionic radius between Ni and Li ions[87,88]. The onset potential of oxygen loss decreases with increasing Ni content, for example, at around 4.6 V

for NMC111 and at 4.2 V for NMC811[88,89]. Therefore, Ni-rich cathodes undergo severe structural degradation at lower charging voltages. To date, surface coating and modification stand out as the predominant methodologies for curbing undesirable side reactions between oxide cathodes and solid electrolytes[13–15]. While irreversible surface degradation plays a role in capacity loss, the significant hurdle arises from crystal defect-induced phase transitions from the layered structure to the spinel-like phase within the interior of cathode particles, posing a formidable barrier to the practical implementation of Ni-rich and Li-rich cathodes[4,5,16,17,30]. In light of this, it becomes imperative to explore additional approaches that can complement established surface engineering methods.

Addressing crystal defects and lattice strain challenges in Ni-rich cathodes necessitates a holistic consideration of the heterogeneous cathode microstructure, and large anisotropic volume changes driven by Li transport. This calls for a fundamental consideration of composition design and microstructure regulation. Our findings underscore the potential of morphological optimisation and a controlled rate of Li (de)intercalation as chemistry-agnostic strategies to enhance stability against oxygen loss. As shown in Fig. 7a, b, reducing the secondary particle diameter to below 4 μm can effectively mitigate the capacity fading issue observed in conventional polycrystalline cathodes[71,90]. Reducing the diameter of secondary particles from 9 μm to below 4 μm enables the increase of the discharge rate from 1 C to 5 C while maintaining the same 90% usable capacity. This improvement is attributed to the effective shortening of the Li intercalation path and a reduced accumulation of crystalline defects, such as dislocations, during battery operation. TEM analysis further supports these findings, suggesting that reducing the charge rate to below 0.1 C or decreasing the particle size to 1 μm can mitigate structural defects and microcracks, thereby enhancing the cycling stability of Ni-rich NMC cathodes[91]. Moreover, the simulation results presented in Figs. 4d and 7b indicate that employing single crystal cathodes can significantly reduce the formation of dislocations and mitigate oxygen-loss-related structural degradation in Ni-rich NMC cathodes. Differential electrochemical mass spectroscopy characterisation and HRTEM characterisation also reveal that the formation of rock-salt phase in the bulk of cathodes is considerably reduced in single crystal Ni-rich cathodes, compared to polycrystalline cathodes[3,92–96]. Therefore, the improved chemomechanical and cycling performance of single crystal NMC cathodes can be attributed to both the elimination of intergranular cracks and the reduction in defects-induced bulk structural degradation. It is noteworthy that, as shown in Fig. 4d, a high discharge rate of 5 C still results in a spatially heterogeneous distribution of Li-ions inside the single crystal cathode particle, which induces a large strain gradient and triggers the accumulation of dislocations at the edges of single crystal particles, despite the absence of intergrain boundaries in these cathodes. As characterised by the multiscale spatial resolution diffraction and imaging experiments[97], these structural defects cannot be eliminated simply with Li-ion deintercalation or reinsertion in NMC single crystal cathodes. The accumulation of these unrecoverable crystal defects through repeated cycling at high charge rates exacerbates irreversible phase transformation, ultimately leading to the electrochemical decay of single crystal cathodes. To mitigate dislocation formation, it is crucial to enhance Li-ion diffusion kinetics and suppress heterogeneous electrochemical reactions. This can be accomplished by reducing the size of such single crystal cathodes, regulating the crystal facets to reduce Li-ion diffusion pathways, modifying the crystal structure to minimise Li or Ni antisite disordering, and improving electronic conductivity[97].

Figure 7c shows that using a single type of solid electrolytes cannot simultaneously mitigate the large tensile stress buildup in NMC cathodes and on the interfaces between these cathodes and the solid

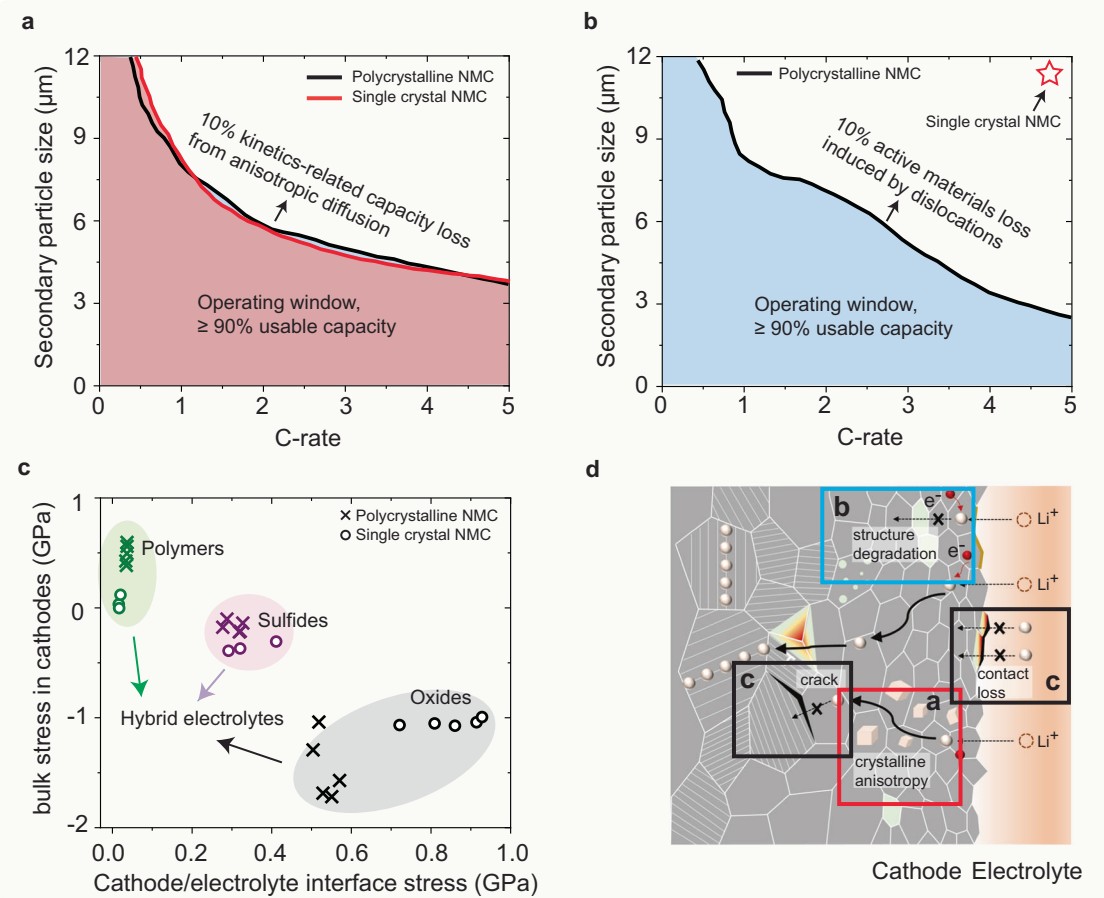

**Fig. 7 | Microstructural effects in solid-state cathode composites. a** Contours of kinetically-induced capacity loss of 0.1 due to anisotropic and concentration-dependent diffusion as a function of particle diameter and C-rate. **b** Contours of 10% loss of active materials induced by the formation of dislocations as a function of particle diameter and C-rate. **c** Relationships between the bulk stress within NMC cathodes and the interface stress (maximum tensile stress) between cathodes and the solid electrolytes. Different points correspond to the results under various discharge rates from 0.25 C to 5 C. **d** Schematic view of different chemo-mechanical degradation mechanisms: kinetically-induced capacity loss; structural degradation induced by dislocations; crack formation and contact loss.

electrolytes. This often leads to either the formation of intergranular cracks within the cathodes or contact loss along the cathode/solid-electrolyte interfaces. Therefore, hybrid systems that involve both oxides and polymer electrolytes seem to be a promising solution to address the mechanical degradation issues in solid-state batteries[1]. Furthermore, preventing cation disordering emerges as a potent means to inhibit the structural changes required to accommodate oxygen vacancies. For instance, the ribbon superstructure, as opposed to the honeycomb superstructure in TM metal layers within sodium-ion intercalation cathodes proves capable of suppressing manganese disorder and the associated oxygen molecule formation, during the P2 to O2 phase transition involving slab gliding[30,98,99].

The large anisotropic volume changes of Ni-rich cathode materials during cycling result in mechanical degradation at the cathode/solid-electrolyte interface, which leads to the increase of electrode impedance and capacity loss[100]. This phenomenon is compounded by the substantial formation of crystal defects within the cathode particles, to reduce the stress magnitude and accommodate the large compositional strain. First-principle calculations suggest that transition-metal centres with non-bonding electronic configurations, cation disordering, redox-inactive species, isotropic structures, and octahedral-to-tetrahedral migration of Li can effectively minimise volume changes upon (dis)charging[101]. For instance, the introduction

of compositionally complex dopants (Ti, Mg, Nb, and Mo) to Ni-rich layered cathodes or the incorporation of a coherent perovskite phase into the layered structure demonstrates a marked reduction in lattice dimension changes over a wide electrochemical window[102,103]. This approach effectively mitigates the structural degradation through a pinning effect. Furthermore, eliminating synthesis-induced crystal defects, such as low-angle tilt boundaries, anti-phase boundaries, stacking faults, and dislocations, contributes to superior structural stability at high voltages while preventing irreversible oxygen release[92,104].

In conclusion, our study presents and applies a mesoscale chemo-mechanical constitutive model for investigating the effect of grain-level chemo-mechanics on the electrochemical performance and degradation mechanisms in composite cathodes of solid-state batteries. Integrating multi-scale experimental and theoretical findings, we reveal that Ni-rich cathodes experience extensive dislocation formation (over 12% plastic shear locally) during discharge, due to the large compositional strains, crystalline anisotropy, and non-equilibrium Li-ion intercalation dynamics. Anisotropic diffusion within the *a-b* plane, coupled with the high Li diffusivity sensitivity to Li content, lead to the heterogeneous Li distribution within cathodes, causing a marked increase in overpotential and capacity loss. Accumulated dislocations on the cathode periphery induce large lattice

strain and profoundly alter the local oxygen environment, potentially triggering oxygen release and the displacement of TM ions into the alkali metal layers. The formation of crystal defects in the bulk of cathode particles driven by Li intercalation can not be alleviated by the established surface coating and modification approaches. Strategies such as reducing the secondary particle diameter to below 4 μm, using single crystal cathodes without interparticle boundaries, and compositionally complex doping, prove effective in countering the generation of crystal defects and addressing capacity fading in conventional polycrystalline cathodes. The results also imply that contact loss at the cathode/solid-electrolyte interface is intricately linked to the anisotropic volume change during Li (de)intercalation and the mechanical properties of solid electrolytes. Reduction in current densities proves insufficient to alleviate the persistent tensile stress and contact loss at the interface. This work provides insights into the role of crystalline anisotropy and grain-level chemo-mechanics in crystal defect generation and mechanical degradation mechanisms of composite electrodes with Ni-rich cathodes, offering valuable contributions to improving energy storage system design.

## Methods

### Governing equations

The free energy of the NMC811 cathode includes the chemical free energy per unit volume of a stress-free homogeneous NMC crystal and the mechanical energy,

$$\psi(c_{Li}, \mathbf{F}) = \psi_{chem}(c_{Li}) + \psi_{mech}(c_{Li}, \mathbf{F}), \tag{1}$$

where $\psi_{chem}$ and $\psi_{mech}$ describe the bulk chemical and mechanical energy density, respectively. The maximum molar density of Li lattice sites is $C_{max} = 49200$ molsm$^{-3}$[56]. $c_{Li}$ is the molar density of Li at a spatial position at a given time. The Li occupancy fraction $\theta$ can then be calculated as $\theta = c_{Li}/C_{max}$. $\mathbf{F}$ is the total deformation gradient.

Li reaction at the cathode/solid-electrolyte interface and diffusion within the cathode is modelled by the Cahn-Hilliard reaction-diffusion equation[46,53],

$$\frac{\partial c_{Li}}{\partial t} = C_{max} \frac{\partial \theta}{\partial t} \\ = -\nabla \cdot \mathbf{j} + R(\theta, \eta), \tag{2}$$

in which $t$ is time, $\mathbf{j}$ is the corresponding Li flux density, and $R$ is the reaction rate depending on the local state-of-charge and the overpotential $\eta$.

Since the relaxation time for mechanical deformation is fast for solids compared to reaction-diffusion of Li-ion, we assume mechanical equilibrium at all times,

$$\nabla \cdot \mathbf{P} = \mathbf{0}, \tag{3}$$

where the left-hand side is the divergence of the first Piola-Kirchhoff stress $\mathbf{P}$.

### Reaction-diffusion model

The interface reaction is described by a boundary condition expressing mass conservation on the reactive surfaces,

$$R = -\mathbf{n} \cdot \mathbf{j}_r, \tag{4}$$

where $\mathbf{j}_r$ is the diffusive flux on the surface, and $\mathbf{n}$ is the unit normal to the reactive surface. The reaction rate depends on the local state-of-charge and the local overpotential, which can be described via generalised Butler-Volmer kinetics[53]. In this study, galvanostatic

simulations are performed for a given (dis)charge rate. Consequently, a constant ionic flux is applied normal to the reactive surface of the secondary particle and is directly related to the current density. Then, at a (dis)charge rate of $n$C, the reaction rate is given by

$$R = nQ\rho \frac{V_{NMC}}{A_{NMC}}, \tag{5}$$

where $Q = 203$ mAhg$^{-1}$ and $\rho = 4.78$ g cm$^{-3}$ are the practical capacity and density of NMC811 cathode[56], respectively. $V_{NMC}$ and $A_{NMC}$ are volume and surface area of the spherical cathode particle. The voltage of the cathode particle is calculated as the average value over the entire surface of the cathode particle.

The regular solution model is used to model the homogeneous chemical free energy,

$$\psi_{chem} = C_{max} \left[ k_B T(\theta \ln\theta + (1-\theta)\ln(1-\theta)) + E_1\theta + E_{11}\theta^2 \right] \tag{6}$$

where $k_B$ is the universal gas constant, $T$ is temperature, $E_1$ and $E_{11}$ are material coefficients that can obtained from the open circuit voltage of a NMC half-cell[18].

The flux force for Li-ion diffusion is described by

$$\mathbf{j} = -C_{max}\mathbf{M} \cdot \nabla\mu, \tag{7}$$

where $\mathbf{M}$ is the anisotropic mobility tensor. The magnitude of the mobility tensor $\mathbf{M}$ is measured by the solid-state nuclear magnetic resonance spectroscopy experiments[18,57]. $\mu$ is Li chemical potential, which contains the chemical part, $\mu_{chem}$, and the mechanical part, $\mu_{mech}$. Lithium intercalation in the cathode is driven by the gradient of the Li chemical potential $\mu$. The chemical potential of Li in NMC cathode is defined as the variational derivative of the total free energy to Li concentration,

$$\mu = \frac{\delta\psi}{\delta c_{Li}} \\ = \frac{\delta\psi_{chem}}{\delta c_{Li}} + \frac{\delta\psi_{mech}}{\delta c_{Li}} \\ = \mu_{chem} + \mu_{mech} \\ = k_B T \ln\frac{\theta}{1-\theta} + E_1 + 2E_{11}\theta + \mu_{mech}. \tag{8}$$

The mechanical contribution in Li chemical potential, $\mu_{mech}$, is described in the following section.

### Mechanical constitutive model

The finite-strain deformation field $\boldsymbol{\chi}(\mathbf{x}) : \mathbf{x} \in \mathcal{B}_0 \rightarrow \mathbf{y} \in \mathcal{B}$ maps material points $\mathbf{x}$ in the reference configuration $\mathcal{B}_0$ to points $\mathbf{y}$ in its deformed configuration $\mathcal{B}$. The total deformation gradient is defined as

$$\mathbf{F} = \frac{\partial\boldsymbol{\chi}}{\partial\mathbf{x}} = \nabla\boldsymbol{\chi}. \tag{9}$$

In the current study, the deformation gradient $\mathbf{F}$ is multiplicatively decomposed as

$$\mathbf{F} = \mathbf{F}_e\mathbf{F}_c\mathbf{F}_p, \tag{10}$$

in terms of the elastic strain $\mathbf{F}_e$, dislocation-induced plastic strain $\mathbf{F}_p$, and compositional eigenstrain $\mathbf{F}_c$[46,47,105]. The compositional eigenstrain $\mathbf{F}_c$ is Li-ion concentration-dependent and links the relation between the local state-of-charge and lattice

mismatch in the *a*, *b,* and *c* directions, respectively. The plastic deformation gradient $\mathbf{F}_P$ is incorporated to depict the accumulation of plastic shear arising from $a - b$ plane dislocations within cathodes.

The local deformation arising from Li-ion (de)intercalation is given by

$$\mathbf{F}_c = \mathbf{I} + \mathbf{V}_c. \tag{11}$$

The components of $\mathbf{V}_c$ can be expressed as:

$$\mathbf{V}_c = \begin{bmatrix} \nu^a & 0 & 0 \\ 0 & \nu^b & 0 \\ 0 & 0 & \nu^c \end{bmatrix}, \tag{12}$$

in the crystal's coordinate system. The (de)lithiation strain $\mathbf{F}_c$ is determined as a function of Li concentration through the experimentally measured lattice parameter changes along different crystallographic directions[19].

The evolution of plastic deformation gradient $\mathbf{F}_p$ is given in terms of the plastic velocity gradient $\mathbf{L}_p$ by the flow rule

$$\dot{\mathbf{F}}_p = \mathbf{L}_p \mathbf{F}_p. \tag{13}$$

To describe the dislocation slip on the $a-b$ plane, a crystal plasticity model[47,48,105] is used, where the plastic velocity gradient $\mathbf{L}_p$ consists of the slip rates $\dot{\gamma}^\alpha$ on crystallographic slip systems,

$$\mathbf{L}_p = \sum_\alpha \dot{\gamma}^\alpha \mathbf{m}^\alpha \otimes \mathbf{n}^\alpha, \tag{14}$$

where $\dot{\gamma}^\alpha$ denotes the shear rate on slip system $\alpha$, and vectors $\mathbf{m}^\alpha$ and $\mathbf{n}^\alpha$ are the slip direction and slip plane normal of slip systems, respectively. The shear rate is given in terms of the phenomenological description as

$$\dot{\gamma}^\alpha = \dot{\gamma}_0 \left| \frac{\tau^\alpha}{g^\alpha} \right|^n \operatorname{sgn}(\tau^\alpha), \tag{15}$$

where $\dot{\gamma}_0$ is the reference shear rate, $\tau^\alpha$ is the resolved shear stress on slip system $\alpha$, $n$ is the strain-rate sensitivity exponent. The slip resistance $g^\alpha$ evolves from its initial value ($g_0$) asymptotically to a system-dependent saturation value $g_\infty$ and depends on shear on slip systems ($\gamma^\beta$, $\beta = 1, 2, 3$) according to the relationship

$$\dot{g}^\alpha = \dot{\gamma}^\beta h_0 |1 - g^\beta / g_\infty|^a \operatorname{sgn}\left(1 - g^\beta / g_\infty\right) h_{\alpha\beta}, \tag{16}$$

with hardening parameters $h_0$, $a$, and $h_{\alpha\beta}$.

The mechanical energy density, $\psi_{\text{mech}}$, is modelled by the following form

$$\psi_{\text{mech}} = \frac{1}{2} \mathbf{E}_e \cdot \mathbb{C} \mathbf{E}_e, \tag{17}$$

where $\mathbb{C}$ is the anisotropic elastic stiffness. $\mathbf{E}_e$ is the Green-Lagrange strain, which is calculated as

$$\mathbf{E}_e = \frac{1}{2} \mathbf{F}_c^T \left( \mathbf{F}_e^T \mathbf{F}_e - \mathbf{I} \right) \mathbf{F}_c, \tag{18}$$

The work conjugate second Piola-Kirchhoff stress $\mathbf{S}$ is given by,

$$\mathbf{S} = \mathbb{C} \mathbf{E}_e, \tag{19}$$

which is related to the first Piola-Kirchhoff stress $\mathbf{P}$ through

$$\mathbf{P} = \mathbf{F}_e \mathbf{F}_c \mathbf{S} \mathbf{F}_p^{-T}. \tag{20}$$

Combination of Eqs. (10), (11), and (17) to (20) then yield the mechanical contribution for the Li chemical potential in Eq. (8) as

$$\begin{aligned} \mu_{\text{mech}} &= \frac{\delta \psi_{\text{mech}}}{\delta c_{\text{Li}}} \\ &= -\frac{1}{C_{\text{max}}} \mathbf{F}_c \mathbf{S} \cdot \frac{\partial \mathbf{V}_c}{\partial \theta}. \end{aligned} \tag{21}$$

### Oxygen vacancy formation energy

The formation of crystal defects, such as dislocations, not only leads to large lattice strain but also dramatically modifies the local oxygen environment, which manifests through inserting extra lattice planes or perturbing the sequence of oxygen layers[4,5,21,22]. The role of lattice strain arsing from dislocations in the formation energy of oxygen vacancies in the layered oxide structure is assessed via atomic-scale calculations[4,22,28,29]. The formation energy of oxygen vacancy ($E_O$) is given by the following equation:

$$E_O[V_O] = E_{\text{total}}[V_O] - E_{\text{total}}[P] + \mu_O, \tag{22}$$

where $E_{\text{total}}[V_O]$ and $E_{\text{total}}[P]$ denote the total energy of the supercells with and without an oxygen vacancy, respectively. $\mu_O$ is the chemical potential of oxygen, where the gas phase $O_2$ molecule is used as the reference state. The details of these first principle calculations can be found in the references[4,22,28,29]. Supplementary Fig. S3 shows the influence of the applied tensile strain on the formation energy of oxygen vacancy in the layered oxide structure. The results indicate that the increased lattice strain notably decreases the energy barrier to remove lattice oxygen[4]. To account for the impact of dislocation-induced shear during Li (de)intercalation on oxygen release in our mesoscale chemo-mechanical simulations, the material domains with a plastic shear exceeding 12% are delineated as the oxygen-deficient phase.

### Numerical implementation

The developed chemo-mechanical model was implemented in the freeware simulation package DAMASK, specifically in DAMASK v2.0.2 version[48]. A large-scale parallel finite element solver using the PETSc numerical library[45] was developed to handle the discretization and numerical solution of the coupled governing equations[46,47].

The weak form of the linear momentum balance equation, Eq. (3), is given by

$$\mathbf{0} = \int_{\mathcal{B}_0} \nabla \delta \boldsymbol{\chi} \cdot \mathbf{P} \, d\mathbf{x}, \tag{23}$$

where $\delta \boldsymbol{\chi}$ is the admissible variation of the deformation field $\boldsymbol{\chi}$. The deformation gradient partitioning (Eq. (10)) and the stresses (Eq. (19)) are calculated based on a two-level iterative predictor-corrector scheme, as described in detail in Shanthraj et al.[72].

The weak form of the reaction-transport relation, Eq. (2), is given by

$$\int_{\mathcal{B}_0} \left[ \left( \dot{\theta} - \frac{1}{C_{\text{max}}} R \right) \delta \mu + M \nabla \mu \cdot \nabla \delta \mu \right] d\mathbf{x} = 0, \tag{24}$$

where $\delta \mu$ is the virtual chemical potential. It is worth noting that Eq. (2) is reformulated as a function of chemical potential rather than the

composition as the independent field variable, using a semi-analytical inversion approach[46,106,107].

The deformation field, $\chi(\mathbf{x})$, and chemical potential, $\mu(\mathbf{x})$, in addition to their virtual counterparts are discretised using a finite element basis of shape functions, $N_i^\chi$, $N_i^\mu$, $N_i^{\delta\chi}$, and $N_i^{\delta\mu}$, where $[\chi]_i$, $[\mu]_i$, $[\delta\chi]_i$, and $[\delta\mu]_i$ are the degrees of freedom, respectively. The corresponding discrete differential operator matrices are $\mathbf{B}_i^\chi$, $\mathbf{B}_i^\mu$, $\mathbf{B}_i^{\delta\chi}$, and $\mathbf{B}_i^{\delta\mu}$. The weak forms of Eqs. (23) and (24) are reformulated as

$$\sum_i [\delta\chi]_i^T \int_{\mathcal{B}_0} \left[\mathbf{B}_i^{\delta\chi}\right]^T \mathbf{P}\,d\mathbf{x} = \sum_i [\delta\chi]_i^T \mathcal{R}_i^{mech} = \mathbf{0}, \qquad (25)$$

$$\sum_i [\delta\mu]_i^T \int_{\mathcal{B}_0} \left[\left[N_i^{\delta\mu}\right]^T \left(\dot\theta - \frac{1}{C_{max}}R\right) + \left[\mathbf{B}_i^{\delta\mu}\right]^T MB_i^\mu[\mu]_i\right] d\mathbf{x}$$
$$= \sum_i [\delta\mu]_i^T \mathcal{R}_i^{chem} = 0, \qquad (26)$$

where $\mathcal{R}_i^{mech}$ and $\mathcal{R}_i^{chem}$ are the residual values at node $i$ of the respective fields. A time-discrete system of equations is obtained by using a backward Euler approximation method

$$\dot\theta = \frac{\theta(t_n) - \theta(t_{n-1})}{\Delta t}. \qquad (27)$$

The self-consistent solution of the coupled fields of Eqs. (25) and (26) are achieved using a staggered iteration method for each time increment. This staggered iteration method allows the solution scheme of each field to be prescribed independently. The mechanical equilibrium equation, Eq. (25), is solved using an inexact Newton method with a critical point secant line search[108]. For the reaction-diffusion equation, Eq. (26), the Newton method with a backtracking line search[109] is employed. Within each Newton iteration for both equations, a flexible GMRES linear solver[110] preconditioned with a smoothed-aggregation algebraic multigrid method[111] is employed for the linear solve. The numerical solution procedure is described in detail in Liu et al.[47] and Shanthraj et al.[46].

### Model parametrization
**Chemical free energy.** The homogeneous chemical free energy can be obtained by integrating the homogeneous Li chemical potential, $\mu_{chem}$, in the NMC811 cathode, which can be measured by the open circuit voltage of a NMC811 half-cell[18,57] according to

$$\mu_{chem} = -V_{OC}F + \mu_{Li}^a, \qquad (28)$$

where $V_{OC}$ is the open circuit voltage, and $F$ is the Faraday constant. $\mu_{Li}^a$ denotes the chemical potential of Li metal in the anode. The reference chemical potential $\mu_{Li}^a$ is a constant since the occupancy of Li in the metal anode is invariant. The open circuit voltage versus Li occupancy profile is measured by the galvanostatic intermittent titration technique (GITT) experiment[18]. The GITT experiment is performed on a cell with an NMC811 cathode and Li metal anode. Therefore, as shown in Supplementary Fig. S2, the chemical-free energy landscape can be extracted by fitting the stress-free chemical potential of Li in cathode with the measured open circuit voltage (Supplementary Table S1)[18].

**Diffusivity.** The anisotropic mobility tensor $\mathbf{M}$ is related to the diffusivity tensor $\mathbf{D}$ via the Einstein relation

$$\mathbf{M} = \frac{\mathbf{D}}{k_B T}. \qquad (29)$$

The layered structure of the NMC cathode restricts Li diffusion solely to the basal plane. The anisotropic diffusivity tensor in the crystallographic coordinate system is given by

$$\mathbf{D} = \begin{bmatrix} D_{Li} & 0 & 0 \\ 0 & D_{Li} & 0 \\ 0 & 0 & 0 \end{bmatrix}, \qquad (30)$$

where $D_{Li}$ is the Li concentration-dependent diffusivity within the basal plane. The diffusion coefficient $D_{Li}$ is obtained from the Li concentration-dependent hop rate of Li ions in NMC by the solid-state nuclear magnetic resonance spectroscopy experiments[18,57]. As shown in Fig. 3a, a smooth interpolation function $D_{Li}(\theta)$ is used to fit the measured diffusivity data points from the NMR experiments (Supplementary Table S1),

$$D_{Li}(\theta) = D_{ref}\,\exp\left(D_0 + D_1\theta + D_2\theta^2 + D_3\theta^3 + D_4\theta^4 + D_5\theta^5\right). \qquad (31)$$

**Lattice dimension change.** The (de)lithiation of NMC811 cathode materials leads to a dilatation and distortion of their atomic structures. The atomic lattice dimension change during charge and discharge is measured by operando synchrotron XRD experiments[8,18,19]. Figure 3b shows the evolution of the lattice strain in the $c$ and $a$ direction of the NMC811 cathode obtained from XRD experiments[18]. The symmetry of the layered oxide crystal structure indicates that the cathode exhibits the same lattice strain in the $a$ and $b$ direction. The $a$ and $b$ lattice parameters increase upon lithiation by maximum 2%. The $c$ lattice parameter exhibits a non-monotonic behaviour that it rapidly increases at the initial stage of discharge and gradually collapses as $\theta > 0.37$. These dimensional changes are attributed to the change of Ni oxidation states, and the modification of the interlayer spacing between the $O^{2-}$ planes. Two smooth interpolation functions $\nu^{a,b}(\theta)$ and $\nu^c(\theta)$ are employed to fit the experimentally measured lattice dimension changes during lithiation (Supplementary Table S1),

$$\nu^{a,b}(\theta) = \nu_0^{a,b} + \nu_1^{a,b}\theta + \nu_2^{a,b}c\theta^2 + \nu_3^{a,b}\theta^3 + \nu_4^{a,b}\theta^4 + \nu_5^{a,b}\theta^5, \qquad (32)$$

and

$$\nu^c(\theta) = \nu_0^c + \nu_1^c\theta + \nu_2^c\theta^2 + \nu_3^c\theta^3 + \nu_4^c\theta^4. \qquad (33)$$

**Elasto-plastic properties.** The anisotropic stiffness matrix of the NMC811 cathodes at the fully lithiated and delithiated state is obtained using environmental nanoindentation experiments and first-principles calculations[58–60]. The observed anisotropic elastic behaviour in layered oxide materials is attributed to the weak bonding along the $c$ axis compared to the $a$ axis[58,59]. The slip strength along the basal plane of NMC811 cathode materials is assessed by an indentation-based test method[112]. Isotropic elastic and elasto-plastic mechanical properties are assumed for solid electrolytes within the composite cathode[73–75,113–116].

All the material parameters are summarised in the Supplementary Table S1.

## Data availability
All data to evaluate the conclusions are present in the manuscript, and the Supplementary Material. Raw data are available from the corresponding authors on request.

## Code availability
The open-source code for DAMASK v2.0.2 version is available at https://github.com/damask-multiphysics/DAMASK/tree/v2.0.2. The specific code used in this study can be accessed at

https://git.damask-multiphysics.org (branch: plasticity_chemo_mechanics, commit: a987e05f8241fb1ef7d8ff913769ea09ac2cd1a3) with the permission of Max-Planck-Institut für Eisenforschung GmbH. Access to the repository requires registration via email at damask-CLA@mpie.de and approval of the Contributor Licence Agreement, available at https://damask-multiphysics.org/_static/DAMASK_CLA.pdf.

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

## Acknowledgements

C.L. acknowledges the discussion with Dr. Yang Bai on the numerical implementation of the model.

## Author contributions

C.L. conceived the study and conducted modelling and data analysis; D.R. advised on data analysis and result interpretation; C.L. drafted the manuscript; C.L., F.R. and D.R. reviewed the manuscript.

## Funding

## Competing interests
The authors declare no competing interests.
