## [Peer Review File · Nature Communications]

Role of grain-level chemo-mechanics in composite cathode degradation of solid-state lithium batteriesREVIEWER COMMENTS

Reviewer #1 (Remarks to the Author):

The primary focus of this paper is the role of grain-level chemo-mechanics in the degradation of composite cathodes in solid-state lithium batteries. It discusses the impact of factors such as crystalline anisotropy, Li diffusion rates, lattice dimension changes, and contact mechanics on the degradation of cathode microstructures. The document also explores the formation of crystal defects, oxygen loss, and structural degradation in composite cathodes, and provides insights into capacity decay and potential strategies for mitigating capacity loss in solid-state batteries. The manuscript considers scientifically interesting and practically important effects to elucidate the effects of alternative crystalline structures and surely forms the basis for a highly relevant paper. However, it would benefit greatly by better explaining and justifying assumptions as well as providing quantitative details about physical properties and model parameters.

Comment 1: The article focuses on the performance of composite cathodes in all-solid-state batteries (ASSB). However, the developed model does not account for phase separation and fracture between the electrode and electrolyte, which is a major degradation mechanism in ASSB. Additionally, as pointed out in the manuscript, the NMC particles have primary-secondary “meatball” structure. The diffusion-induced stress results in intra and inter-granular gap/crack formation, which is neglected in this work. Numerous studies in the literature have integrated these effects alongside anisotropic behavior, which should be acknowledged [1-6]. Chemo-mechanical degradation must be considered for a more accurate representative model, or at the very least, the assumptions and shortcomings must be clearly stated.

Comment 2: Based on the most experimental work in the literature the single crystal NMC particles outperform the polycrystalline NMC [7-9]. This superiority is often attributed to the elimination of intergranular fracture in single crystal NMC [10]. However, Figure 3-D

presents results that seemingly contradict these established findings. The authors should provide a justification for this inconsistency.

Comment 3: The assumption for the stress-free state is unclear in this work. It must be noted that most of degradation and fracture usually occurs at the end of charge, typically after approximately 0.25 intercalation fraction, where a phase transformation occurs [11]. It is challenging to understand how the models, which focuses on discharge predict this behavior and it's really concerning.

Comment 4: In the main manuscript, the authors meticulously address the impact of anisotropic and concentration-dependent diffusion coefficients on NMC particles. However, the anisotropic and concentration-dependent mechanical properties, such as stiffness parameters, and chemical expansion coefficients appear to be overlooked unless the reader refers to the supplementary document. It's crucial to emphasize that these anisotropic properties significantly change the stress distribution and related results. This demands clear acknowledgment, comparison and discussion within the main manuscript.

[1] R. Xu, Y. Yang, F. Yin, P. Liu, P. Cloetens, Y. Liu, F. Lin, and K. Zhao.

Heterogeneous damage in Li-ion batteries: Experimental analysis and theoretical modeling. *J. Mech. Phys. Solids*, 129:160–183, 2019.

[2] Y. Bai, K. Zhao, Y. Liu, P. Stein, and B.-X. Xu, A chemo-mechanical grain boundary model and its application to understand the damage of Li-ion battery materials, *..*,183:45–49, 2020.

[3] K. Taghikhani, P.J. Weddle, J.R. Berger, R.J. Kee, Modeling Coupled

Chemo-Mechanical Behavior of Randomly Oriented NMC811 Polycrystalline Li-Ion Battery Cathodes, *J. Electrochem. Soc.* 168:080511, 2021.

[4] K. Taghikhani, P.J. Weddle, R.M. Hoffman, J.R. Berger, R.J. Kee, Electro-chemo-mechanical finite-element model of single-crystal and polycrystalline NMC cathode particles embedded in an argyrodite solid electrolyte. *Electrochim. Acta.*, 460:142585, 2023.

[5] J.M. Allen, P.J. Weddle, A. Verma, A. Mallarapu, F. Usseglio-Viretta, D.P. Finegan, A.M. Colclasure, W. Mai, V. Schmidt, O. Furat, D. Diercks, T. Tanim, and K. Smith, Quantifying the influence of charge rate and cathode-particle architectures on degradation of li-ion cells through 3D continuum-level damage models, *J. Power Sources*, 512:230415, 2021.

[6] P. Barai, Computational elucidation of mechanical degradation in NMC cathodes: Impact on cell performance, *J. Electrochem. Energy Convers. Storage*, 19(4):041007, 2022.

[7] J. Li, A.R. Cameron, H. Li, S. Glazier, D. Xiong, M. Chatzidakis, J. Allen, G.A. Botton, and J.R. Dahn. Comparison of Single Crystal and Polycrystalline $\text{LiNi}_{0.5}\text{Mn}_{0.3}\text{Co}_{0.2}\text{O}_2$ Positive Electrode Materials for High Voltage Li-Ion Cells, *J. Electrochem. Soc.*, 164(7): A1534-A1544, 2017.

[8] G. Qian, Y. Zhang, L. Li, R. Zhang, J. Xu, Z. Cheng, S. Xie, H. Wang, Q. Rao, Y. He, et al., Single-crystal nickel-rich layered-oxide battery cathode materials: synthesis, electrochemistry, and intra-granular fracture, *Energy Storage Mater.* 27: 140–149, 2020.

[9] S. Payandeh, D. Goonetilleke, M. Bianchini, J. Janek, and T. Brezesinski, Single versus poly-crystalline layered oxide cathode materials for solid-state battery applications - a short review article, *Cur. Opin. Electroch.*, 31:100877, 2022.

[10] R. Ruess, S. Schweidler, H. Hemmelmann, G. Conforto, A. Bielefeld, D.A. Weber, J. Sann, M.T. Elm, and J. Janek, Influence of NCM particle cracking on kinetics of lithium-ion batteries with liquid or solid electrolyte, *J. Electrochem. Soc.*, 167 (10):100532, 2020.

[11] Y. Han, S.H. Jung, H. Kwak, S. Jun, H.H. Kwak, J. H. Lee, ... and Y. S. Jung. Single-or poly-crystalline Ni-rich layered cathode, sulfide or halide solid electrolyte: which will be the winners for all-solid-state batteries?, *Advanced Energy Materials*, 11:2100126, 2021.

Reviewer #2 (Remarks to the Author):

In this work, Liu et al. proposed a microstructure-sensitive chemo-mechanical model to study the degradation of NMC cathode in solid-state lithium batteries. Some interesting topics are discussed in detail, particularly the impact of dislocations on structural degradation and capacity loss of the battery material from theoretical and numerical points of view. There is no doubt this work extended the understanding of the degradation

mechanism in electrode materials at the microstructure level. However, due to the limitations of theoretical and numerical models in general, a few issues must be addressed before the manuscript reaches the standard of Nature Communications and can be further considered for publication:

1. The author presents a coupled chemo-mechanical crystal plasticity model for lithium-ion batteries. However, there have been similar models for coupled chemo-mechanical problems, e.g. [Journal of the Mechanics and Physics of Solids, 2023, 173: 105190.] and [International Journal of Plasticity, 2024, 173: 103890.]. The author should emphasize the novelty of the model. Otherwise, this is simply an application of the model for electrode materials of lithium-ion batteries.

2. While grain boundaries in the polycrystal particles are crucial for the performance and degradation of the materials, it is not discussed or properly modeled in the present theoretical model. How do grain boundaries influence the predictions for polycrystals in this work?

3. There is a lack of experimental results to support the numerical predictions. Since the critical size of the secondary particle diameter has been quantitatively predicted to be 4 micrometers, are there any experimental results that can at least partly support this prediction? Similarly, are there any experimental results that can partly support the prediction for capacity loss?

4. The continuum crystal plasticity model does not consider the self-stress of dislocations, which can significantly influence local concentrations and lead to diffusion heterogeneity, the author should discuss these influences, as well as the limitation of the model.

5. Why the dislocation activity is more significant in polycrystals? Since polycrystals consist of smaller primary particles, it is believed that the formation of misfit dislocations in primary particles in smaller sizes is less favorable from an energy point of view [International Journal of Mechanical Sciences, 2020, 183: 105812.]. Some discussion on the numerical prediction must be made after presenting these results.

6. It is important to understand how dislocations cause degradation of the material, but recent studies show dislocations are also an excellent tool to improve the performance of the battery materials [Journal of the American Chemical Society, 2024, 146(2): 1710-1721.]. Does the author's model also predict similar effects of dislocations for the NMC electrode materials?

7. While this manuscript is based on theoretical models, the model details are very limited, e.g. the explicit formulation of chemical potential is not given in Eq. (7). There are also no details of the numerical models. Without these details, it can be difficult to follow for potential readers. I would ask the author to provide the details of the model.

8. The author should provide the declaration for all quantities presented in the manuscript, for example, what are the definitions of the normalized and kinetic capacity loss in the numerical model?

9. What is the numerical method for solving the governing equations and the numerical model is implemented in which software?

10. In Fig. 3, the symbol P is used for the first Piola-Kirchhoff stress in the stress equilibrium, but the symbol σ is used for Cauchy stress in the free energy which is not declared in the Methods. This is very confusing to the reviewer regarding the numerical implementation of the theoretical model.

11. In Eq. (7), the variation should be with respect to physical quantities rather than a dimensionless quantity c ?

Reviewer #3 (Remarks to the Author):

Authors developed chemo-mechanical model to understand the degradation of polycrystalline NMC-811 particle embedded in solid electrolyte. They utilized crystal plasticity theory to simulate the elastoplastic behavior of cathode particles; however, the solid electrolyte is purely elastic. Secondary particle size, C-rates and elastic modulus of SE are varied to understand their effect on the chemo-mechanical behavior of polycrystalline NMC-811 cathode particle. Even though the formulation has novelty of having crystal plasticity and oxygen loss dynamics, the results about using small particle size and low C-rates are useful in mitigating the capacity loss are very well reported in the literature. To improve the quality of the manuscripts, kindly consider addressing following comments. There are also some major flaws in understanding the literature which are also mentioned below.

- The authors mentioned in Page 2, line 12 that volume change for NMC-811 is 7.8% without any reference. In the article by Yoon et al., 2006 (<https://www.sciencedirect.com/science/article/pii/S138824810600227X>), the volume change for NMC-811 was measured to be ~3.4%. Can authors please comment on how they found 7.8% volume expansion?

- About the authors comment on Page 2, line 22, “Although the exact nature of the defect-induced bonding environment change is debated and may vary among different materials, such as Ni-rich and Li-rich layered oxides, the large lattice strain associated with crystal defects can markedly reduce the energy barriers to remove lattice oxygen and trigger Li/TM ion mixing [4, 20, 26–29]. Therefore, besides the electrochemical instability of the cathode/electrolyte interface, the continuous accumulation of crystal defects in cathode particles due to Li (de)intercalation over cycling acts as the primary driving force for aggravating oxygen loss and altering the structural stability.” Oxygen loss and structural change mostly happens if voltages > 4.2 (see below review article for the detail). Can authors please clarify if oxygen loss and structural change is significant enough within the operating voltage (i.e., <4.2)?

<https://onlinelibrary.wiley.com/doi/full/10.1002/aenm.202103712>

- About the author’s statement on Page 2 and line 40. “However, the assessment of how microstructure and (dis)charging protocols affect micromechanics and the formation of crystal defects in composite cathodes remains unexplored. This gap in knowledge is primarily due to the limitations and efforts associated with the application of advanced and

standardized analytical techniques in probing cathodes and their environments across various length scales [4, 5].” Can author clarify this literature gap as there are already several papers listed below understanding the chemo-mechanical behavior of NMC particles with or without solid electrolyte for different C-rates, grain sizes and interface energies?

<https://www.sciencedirect.com/science/article/pii/S0378775321009216>

<https://www.sciencedirect.com/science/article/pii/S0378775323014106#sec3>

<https://www.sciencedirect.com/science/article/pii/S0378775324000053>

<https://www.sciencedirect.com/science/article/pii/S0022509622000540>

<https://www.sciencedirect.com/science/article/pii/S0022509621002556>

- The author’s statement on Page 4 and line 2 is wrong. “As 2 characterized by electron backscatter diffraction measurements [30], the NMC811 3 polycrystalline secondary particles exhibit randomly distributed crystallographic orientations (Fig. 3b).” Actually, NMC-811 has radially oriented grains and in the reference provided by the authors, the same has been mentioned. The reviewer advised authors to carefully read the literature before making such statements. With this statement being wrong, the authors need to revise all the results provided in the paper.

- As authors only considered isotropic elastic deformation for the solid electrolyte. Can authors please comment on it as solid electrolyte also shows elasto-plastic and rate dependent behavior as reported by Papakyriakou et al., J. Power Sources (2021) and Athanasiou et al., Cell Report Phys. Science (2022)?

- On page 4, line 25, the authors mentioned that “The material domains 25 with a plastic shear exceeding 12% in cathode particles after discharge are categorized 26 as the oxygen-deficient phase.” Can authors please clarify how such number is decided?

- Authors simulated three types of simulations i.e., isotropic, anisotropic Li diffusion inside the primary particle with a constant Li diffusivity, and anisotropic diffusion with a Li concentration-dependent diffusion coefficient. Can author please clarify the intension behind simulating such cases as only last one is the practical case?

- Can authors please clarify how they measure the single voltage profile for each case, as for anisotropic particle the voltage on each point of particle surface will be different?

- Better insights can be obtained from studying the actual reconstructed microstructure? Authors should consider using the reconstructed microstructure.

- Authors statement on Page 6, line 7 “Fig. 3b, c show that the large anisotropic volume change of Ni-rich layered cathodes driven by Li intercalation and the cathode microstructure heterogeneity result in substantial differences in dislocation activity and accumulation between primary particles” is unclear. As mentioned earlier by the reviewer that the volume change for NMC-811 is ~3.5%. Please comment on how such a deformation can be considered as large deformation and from Fig. 3, it is difficult to anticipate that the numbers correspond to large deformation.

- Rather than just showing max. principal stress as in Fig. 3(h), the reviewer advised to show the contour plots of von-Mises stresses.

- Section 4, page 7, the author’s observation that by using small secondary particles and lower C-rates, the capacity can be improved. Such results are obvious and have already been reported by numerous papers. Can authors please comment on the novelty of the results?

- Concern on the results presented by authors for different modulus of solid electrolytes. As shown in Fig. 4(f), the stresses are of the order of 1.6 GPa, can material bear such stresses in reality? Please comment.

- The author’s statement on Page 10, line 3 “Fig. 5c consistently indicate that, on average, primary particles located near the exterior of the agglomerate are more susceptible to oxygen loss compared to those in the interior.” When the voltage goes beyond 4.2 for NMC-811, surface reconstruction happens and there is oxygen loss in the process. Such

results have been reported by several experimental articles. Can the author please comment on the novelty of the results?

- On page 11, line 6, the authors found that particle size greater than 8 μm in diameter tend to exhibit significant capacity loss. Can authors please comment on any experimental evidence or if they performed any experimental investigation to validate such a case?

- On page 11, line 36, "As shown in Fig. 6a,b, strategies including reducing the secondary particle diameter to below 4 μm and employing single crystal layered cathodes devoid of interparticle boundaries have proven effective in mitigating the capacity fading issue observed in conventional polycrystalline cathodes [6, 41]." It has been reported that using single crystal mitigate the fracture within the particle and can reduce capacity loss. Can authors please clarify the novelty of the results and if they have done any experimental investigation on validating that below 4 μm mitigate the capacity fading issue.

Responses to reviewers' comments on "Role of grain-level chemo-mechanics in composite cathode degradation of solid-state lithium batteries"

Chuanlai Liu, Franz Roters, Dierk Raabe

We appreciate the reviewers for their detailed review and constructive comments. Taking the reviewers' feedback on-board, we have significantly enhanced the manuscript by incorporating several new results. Additionally, we have expanded the manuscript to include a comprehensive description of the chemo-mechanical model, detailed model parametrization, and comparisons with experimental results. We believe these revisions have substantially improved the manuscript. Changes are highlighted in orange in the revised manuscript, and responses to the comments are outlined in detail below. To maintain clarity and conciseness, we have followed the new sequence for figures, equations, sections, and page numbers in the revised manuscript.

Reviewer 1

Opening comment: *The primary focus of this paper is the role of grain-level chemo-mechanics in the degradation of composite cathodes in solid-state lithium batteries. It discusses the impact of factors such as crystalline anisotropy, Li diffusion rates, lattice dimension changes, and contact mechanics on the degradation of cathode microstructures. The document also explores the formation of crystal defects, oxygen loss, and structural degradation in composite cathodes, and provides insights into capacity decay and potential strategies for mitigating capacity loss in solid-state batteries. The manuscript considers scientifically interesting and practically important effects to elucidate the effects of alternative crystalline structures and surely forms the basis for a highly relevant paper. However, it would benefit greatly by better explaining and justifying assumptions as well as providing quantitative details about physical properties and model parameters.*

Response: We thank the reviewer for the positive feedback and pertinent comments on the manuscript. We have comprehensively revised the manuscript by including a detailed description of the chemo-mechanical model, model parametrization, and comparisons with experimental results. These additions address the assumptions and provide quantitative details about the physical properties and model parameters.

Comment 1: *The article focuses on the performance of composite cathodes in all-solid-state batteries (ASSB). However, the developed model does not account for phase separation*

and fracture between the electrode and electrolyte, which is a major degradation mechanism in ASSB. Additionally, as pointed out in the manuscript, the NMC particles have primary-secondary “meatball” structure. The diffusion-induced stress results in intra and inter-granular gap/crack formation, which is neglected in this work. Numerous studies in the literature have integrated these effects alongside anisotropic behavior, which should be acknowledged [1-6]. Chemo-mechanical degradation must be considered for a more accurate representative model, or at the very least, the assumptions and shortcomings must be clearly stated.

[1] R. Xu, Y. Yang, F. Yin, P. Liu, P. Cloetens, Y. Liu, F. Lin, K. Zhao. *Heterogeneous damage in Li-ion batteries: Experimental analysis and theoretical modeling*. *J. Mech. Phys. Solids*, 129: 160–183, 2019.

[2] Y. Bai, K. Zhao, Y. Liu, P. Stein, B.-X. Xu. *A chemo-mechanical grain boundary model and its application to understand the damage of Li-ion battery materials*. *Scr. Mater.* 183 (2020): 45-49.

[3] K. Taghikhani, P.J. Weddle, J.R. Berger, R.J. Kee. *Modeling coupled chemo-mechanical behavior of randomly oriented NMC811 polycrystalline Li-ion battery cathodes*. *J. Electrochem. Soc.* 168:080511, 2021.

[4] K. Taghikhani, P.J. Weddle, R.M. Hoffman, J.R. Berger, R.J. Kee. *Electro-chemo-mechanical finite-element model of single-crystal and polycrystalline NMC cathode particles embedded in an argyrodite solid electrolyte*. *Electrochim. Acta.*, 460: 142585, 2023.

[5] J.M. Allen, P.J. Weddle, A. Verma, A. Mallarapu, F. Usseglio-Viretta, D.P. Finegan, A.M. Colclasure, W. Mai, V. Schmidt, O. Furat, D. Diercks, T. Tanim, K. Smith. *Quantifying the influence of charge rate and cathode-particle architectures on degradation of Li-ion cells through 3D continuum-level damage models*. *J. Power Sources*, 512: 230415, 2021.

[6] P. Barai, *Computational elucidation of mechanical degradation in NMC cathodes: Impact on cell performance*. *J. Electrochem. Energy Convers. Storage*, 19(4):041007, 2022.

Response: We thank the reviewer for pointing out the omission of mechanical fracture in the present chemo-mechanical model. While the present study uses maximum principal stress distribution to analyse the associated contact mechanics problems, the current model does not explicitly account for crack formation. We have now acknowledged this limitation in the *Section Theoretical framework and model setup* of the revised manuscript (Below; See Page 4, line 41-43 in the manuscript). Additionally, in the new *Section Preliminary insights and perspectives on model development*, we emphasized the need for future work to integrate mechanical fracture to enhance the description of the chemo-mechanical behaviour in solid-state batteries, and discussed existing models that account for these effects (Below; See Page 13, line 18-41).

“While the present study uses the maximum principal stress distribution to analyse contact mechanics problems, the current model does not explicitly account for mechanical fracture.”

“The current model simplifies or neglects a few other factors that should be addressed to provide a full description of chemo-mechanics in solid-state batteries in the future, for example, explicit description of interfacial and intergranular crack formation [32–35, 38–40, 43, 44, 72], concentration-dependent elastic modulus of NMC cathodes [58–60], plastic deformation in solid electrolytes [58–60], role of discrete dislocations and grain boundaries in Li diffusion and chemo-mechanics of NMC cathodes [33, 35, 76, 77], and generation of the representative electrode particle microstructure from microscopy data [43, 78].”

“Crack formation in composite cathodes. One of the crucial aspects of composite cathodes in solid-state batteries is the stress response of their microstructure to lattice dimension changes induced by Li transport. As shown in Fig. 3, high tensile stresses build up along grain boundaries in cathodes and at the interface between cathodes and solid electrolytes upon electrochemical cycling. This high stress can lead to contact mechanics problems. Loss of interfacial contact within the cell obstructs Li transport in cathodes and charge-transfer reactions at the cathode/electrolyte interface, which subsequently results in resistance increase, capacity loss, and rate performance deterioration in batteries. While this study effectively captures the heterogeneous stress response and crystalline defect formation induced by compositional strains, it does not account for crack formation. Including mechanical fracture in the model would be beneficial to achieve a comprehensive understanding of the mechanical behaviour in solid-state batteries. Particle fracture and interfacial contact loss in electrode materials can be integrated into the chemo-mechanical models using cohesive zone models [32–35], spring analogy approaches [38, 39], continuum-damage models [40], and phase-field damage models [43, 44, 72].”

Comment 2: *Based on the most experimental work in the literature the single crystal NMC particles outperform the polycrystalline NMC [7-9]. This superiority is often attributed to the elimination of intergranular fracture in single crystal NMC [10]. However, Figure 3-D presents results that seemingly contradict these established findings. The authors should provide a justification for this inconsistency.*

[7] J. Li, A.R. Cameron, H. Li, S. Glazier, D. Xiong, M. Chatzidakis, J. Allen, G.A. Botton, and J.R. Dahn. Comparison of single crystal and polycrystalline $\text{LiNi}_{0.5}\text{Mn}_{0.3}\text{Co}_{0.2}\text{O}_2$ positive electrode materials for high voltage Li-ion cells, *J. Electrochem. Soc.*, 164 (7): A1534-A1544, 2017.

[8] G. Qian, Y. Zhang, L. Li, R. Zhang, J. Xu, Z. Cheng, S. Xie, H. Wang, Q. Rao, Y. He, et al., Single-crystal nickel-rich layered-oxide battery cathode materials: synthesis, electrochemistry, and intra-granular fracture, *Energy Storage Mater.* 27: 140–149, 2020.

[9] S. Payandeh, D. Goonetilleke, M. Bianchini, J. Janek, and T. Brezesinski, Single versus poly-crystalline layered oxide cathode materials for solid-state battery applications - a short review article, *Cur. Opin. Electroch.*, 31:100877, 2022.

[10] R. Ruess, S. Schweidler, H. Hemmelmann, G. Conforto, A. Bielefeld, D.A. Weber, J.

Sann, M.T. Elm, and J. Janek, Influence of NCM particle cracking on kinetics of lithium-ion batteries with liquid or solid electrolyte, J. Electrochem. Soc., 167 (10):100532, 2020.

Response: We apologize for the lack of clarity in the original manuscript regarding Fig. 3, where the results for single crystal NMC cathode were not included. The comparisons in the original figure focused on polycrystalline cathode particles with different constitutive features under various scenarios: isotropic diffusivity, anisotropic and concentration-independent diffusivity, and anisotropic and concentration-dependent diffusivity. This might have led to misunderstandings about the presented results.

To address this, we have revised Fig. 3 (Page 7) to include comparisons of the chemo-mechanical behaviour in a cathode particle with isotropic and concentration-independent material properties, as well as single crystal and polycrystal particles with anisotropic and Li concentration-dependent material properties. As shown in Fig. 3h in the revised manuscript, the predicted capacity at 1 C for the single crystal cathode case was around 6 % higher than for a polycrystal cathode. However, this capacity difference only accounts for kinetic diffusion in single crystal and polycrystal cathodes and does not consider the effect of mechanical fracture.

Additionally, in the *Section Role of microstructure in rate performance and defect heterogeneity* (Page 9, line 3-11; Page 10, line 11-18) and *Section Discussion* (Page 17, line 39-44; Page 18, line 1-7), we have systematically compared dislocation formation and defect-induced rock-salt phase formation in single crystal and polycrystal cathodes. It was found that the improved chemo-mechanical and cycling performance of single crystal NMC cathodes can be attributed to the reduction in defects-induced bulk structural degradation, in addition to the elimination of intergranular cracks.

Comment 3: *The assumption for the stress-free state is unclear in this work. It must be noted that most of degradation and fracture usually occurs at the end of charge, typically after approximately 0.25 intercalation fraction, where a phase transformation occurs [11]. It is challenging to understand how the models, which focuses on discharge predict this behavior and it's really concerning.*

[11] Y. Han, S.H. Jung, H. Kwak, S. Jun, H.H. Kwak, J. H. Lee, ... and Y. S. Jung. Single-or poly-crystalline Ni-rich layered cathode, sulfide or halide solid electrolyte: which will be the winners for all-solid-state batteries?, Advanced Energy Materials, 11:2100126, 2021.

Response: We have now defined the stress-free state in the *Section Theoretical framework and model setup* (Page 4, line 33-34) of the revised manuscript. For the separate discharge and charge simulations, a Li occupancy fraction of 0.1 or 0.9 in the NMC cathodes is taken as the stress-free state, respectively.

We have also added new results comparing the separate discharge and charge processes in the revised manuscript (Page 7, Fig. 3f, g; Page 8, line 23-32). Our findings indicate that while the cathode particle exhibits volume expansion under discharge, most regions within the cathode/solid-electrolyte interface experience substantial tensile stress

due to the anisotropic chemical expansion of primary particles, for both single crystal and polycrystal cathodes. Under charging conditions, the anisotropic deformation of primary particles results in high stresses both at the cathode/solid-electrolyte interface and grain boundaries within the polycrystal cathode particle.

Comment 4: *In the main manuscript, the authors meticulously address the impact of anisotropic and concentration-dependent diffusion coefficients on NMC particles. However, the anisotropic and concentration-dependent mechanical properties, such as stiffness parameters, and chemical expansion coefficients appear to be overlooked unless the reader refers to the supplementary document. It's crucial to emphasize that these anisotropic properties significantly change the stress distribution and related results. This demands clear acknowledgment, comparison and discussion within the main manuscript.*

Response: We fully agree with the reviewer's suggestion to emphasize the anisotropic and concentration-dependent mechanical properties in the main manuscript. We have revised Fig. 3 (Page 7) and added a new Fig. 6 (Page 15) to present the effect of the anisotropic and concentration-dependent compositional strains and elastic modulus. We modified the title of the *Section Crystalline anisotropy and kinetics for Li intercalation* to *Section Role of anisotropic and concentration-dependent electrochemical and mechanical properties* (Page 6). The revised manuscript (Page 6, line 22-34; Page 8, line 18-32; Page 13, line 42-44; Page 14, line 1-11) now includes a detailed discussion on the effect of mechanical properties on the chemo-mechanical performance of composite cathodes in solid-state batteries.

Reviewer 2

Opening comment: *In this work, Liu et al. proposed a microstructure-sensitive chemo-mechanical model to study the degradation of NMC cathode in solid-state lithium batteries. Some interesting topics are discussed in detail, particularly the impact of dislocations on structural degradation and capacity loss of the battery material from theoretical and numerical points of view. There is no doubt this work extended the understanding of the degradation mechanism in electrode materials at the microstructure level. However, due to the limitations of theoretical and numerical models in general, a few issues must be addressed before the manuscript reaches the standard of Nature Communications and can be further considered for publication.*

Response: We are most grateful for the kind acknowledgement of our research and its significance in this field. We have significantly revised the manuscript according to the reviewer's suggestions and comments.

Comment 1: *The author presents a coupled chemo-mechanical crystal plasticity model for lithium-ion batteries. However, there have been similar models for coupled chemo-mechanical problems, e.g. [Journal of the Mechanics and Physics of Solids, 2023, 173: 105190.] and [International Journal of Plasticity, 2024, 173: 103890.]. The author should emphasize the novelty of the model. Otherwise, this is simply an application of the model for electrode materials of lithium-ion batteries.*

Response: We thank the reviewer’s insight regarding the originality of our chemo-mechanical crystal plasticity model. In comparison with the existing models mentioned in the references, our model presents several novel features. The model in the first reference (Journal of the Mechanics and Physics of Solids, 2023, 173: 105190) is developed in the small strain framework and solved using Fourier transform methods. The model in the second reference (International Journal of Plasticity, 2024, 173: 103890) is derived in the finite strain framework but implemented in a commercial finite element solver. Our thermodynamically consistent chemo-mechanical model is derived in the finite strain framework, which is suitable for addressing the large volume changes in Ni-rich NMC cathodes. The model incorporates anisotropic and concentration-dependent material properties, providing a more accurate representation of the composite cathode behaviour. Additionally, an in-house large-scale parallel finite element solver, using the PETSc numerical library, was developed to discretize the model and efficiently solve the coupled governing equations. The model is implemented in the freeware simulation package DAMASK. We have modified the *Section Theoretical framework and model setup* to make the originality clearer (Page 4, line 5-13).

Furthermore, we have expanded the manuscript to include a comprehensive description of the chemo-mechanical model, numerical implementation, and model parametrization in the *Section Methods*.

Comment 2: *While grain boundaries in the polycrystal particles are crucial for the performance and degradation of the materials, it is not discussed or properly modeled in the present theoretical model. How do grain boundaries influence the predictions for polycrystals in this work?*

Response: We acknowledge the importance of grain boundaries in influencing the performance and degradation of the polycrystalline cathode materials. In the revised manuscript, we have included a discussion on the role of grain boundaries and existing chemo-mechanical models that explicitly consider grain boundary properties. We have also added new simulation results that explicitly consider the effect of Li transport along grain boundaries on the chemo-mechanical behaviour of NMC cathodes (Below; See Page 15, Fig. 6; Page 16, line 1-33).

“Grain boundaries in polycrystalline cathodes. Grain boundaries, with their distinct physicochemical properties compared to the bulk, noticeably affect Li transport, mechanical failure, and structural degradation in polycrystalline NMC cathodes [14, 20, 76, 82, 83]. For example, the low diffusion barriers of TM ions along grain boundaries can promote TM ion dissolution and accelerate formation of rock-salt phases [14, 20, 83]. Additionally, the atomistic structures of various grain boundaries, such as local structural distortion and charge redistribution, can significantly influence Li transport kinetics [76, 82]. Conductive atomic force microscopy characterisation shows that grain boundaries can generally act as fast pathways for Li transport in LiCoO₂ [82]. First-principles calculations reveal that the coherent $\Sigma 2$ grain boundaries enhance Li migration

both along and across the grain boundary plane, whereas the asymmetric $\Sigma 3$ grain boundaries substantially impede Li transport in NMC cathodes [76]. The role of grain boundaries in Li transport kinetics can be incorporated into the chemo-mechanical model by considering the chemical potential or concentration jumps across grain boundaries [33, 35, 77].

In this study, grain boundaries are represented by two layers of elements located between adjacent grains, as shown in Supplementary Fig. S1. Li diffusivity along grain boundaries is assumed to range from 0.05 to 20 times that in the bulk. As shown in Fig. 6g, i, fast Li transport along grain boundaries can significantly enhance the homogeneity of Li distribution within the polycrystalline cathode particle, and effectively decrease the over-potential on the surface of cathodes. Fig. 6h shows the voltage-capacity profiles at a discharge rate of 1 C, with various Li transport kinetics along grain boundaries. A fivefold increase in Li diffusivity along grain boundaries results in a 20% increase in capacity, while a corresponding fivefold decrease in Li diffusivity leads to an 8% capacity fade. This indicates that, besides the mechanisms intrinsic to the crystal and electronic structure of NMC cathodes, tailoring and optimizing grain boundary properties in polycrystalline NMC cathodes can significantly enhance their rate capacity. For example, solid-state electrolyte infused along the grain boundaries in Ni-rich NMC particles acts as fast channel for Li transport, which realises an increase in capacity retention from 79% to 91.6% after 200 cycles [84]. Fig. 6g, i, j show that high tensile stresses exist at the interfaces between cathodes and solid electrolytes regardless of Li transport kinetics along grain boundaries, which is largely determined by anisotropic compositional strains and mechanical properties of composite cathodes.”

Comment 3: *There is a lack of experimental results to support the numerical predictions. Since the critical size of the secondary particle diameter has been quantitatively predicted to be 4 micrometers, are there any experimental results that can at least partly support this prediction? Similarly, are there any experimental results that can partly support the prediction for capacity loss?*

Response: We agree with the reviewer on the importance of supporting our numerical predictions with experimental data. The experimental analysis of active cathode material loss generally involves the formation of rock-salt phase and isolated cathode materials induced by intergranular fracture. Since this study does not explicitly consider crack formation, only qualitative comparisons can be made between predictions and corresponding experimental characterisation studies. In the *Section Dislocation-induced structural degradation and capacity loss* and *Section Discussion*, we have now included relevant experimental results that corroborate our predictions regarding the effect of the size of secondary particles and charging rate on capacity loss (Below; See Page 12, Fig. 5e; Page 13, line 1-15; Page 17, line 39-44; Page 18, line 1-7).

“In addition, Fig. 5e includes the effect of cathode particle size and cycling conditions on the loss of active cathode material from experimental characterisation [67–71]. It is noteworthy that the experimental analysis of active cathode material loss generally involves the formation of rock-salt phase and isolated cathode materials induced by intergranular fracture [67–71]. Since this study does not explicitly consider crack formation, only qualitative comparisons are made between predictions and experimental characterisations. Fig. 5e shows that the total loss of active cathode material increases with higher charging rate according to experimental characterisation [67–70], which aligns with the predicted results. For example, increasing the charging rate from 0.5 C to 6 C leads to an increase in the loss of active material from 8.5% to 16% [67–70]. Furthermore, experimental characterisation reveals that the capacity fading of cells using small cathode particles (average diameter of 11 μm) can be reduced from 22.1% to 16.6%, compared to those with large cathode particles (16 μm) [71]. This capacity fading mechanism is generally attributed to the loss of Li inventory, plating of Li, and loss of active material from both electrodes.”

“TEM analysis further supports these findings, suggesting that reducing the charge rate to below 0.1 C or decreasing the particle size to 1 μm can mitigate structural defects and microcracks, thereby enhancing the cycling stability of Ni-rich NMC cathodes [91]. Moreover, the simulation results presented in Fig. 4d and Fig. 7b indicate that employing single crystal cathodes can significantly reduce the formation of dislocations and mitigate oxygen-loss-related structural degradation in Ni-rich NMC cathodes. Differential electrochemical mass spectroscopy characterisation and HRTEM characterisation also reveal that the formation of rock-salt phase in the bulk of cathodes is considerably reduced in single crystal Ni-rich cathodes, compared to polycrystalline cathodes [3, 92–96]. Therefore, the improved chemomechanical and cycling performance of single crystal NMC cathodes can be attributed to both the elimination of intergranular cracks and the reduction in defects-induced bulk structural degradation.”

Comment 4: *The continuum crystal plasticity model does not consider the self-stress of dislocations, which can significantly influence local concentrations and lead to diffusion heterogeneity, the author should discuss these influences, as well as the limitation of the model.*

Response: We thank the reviewer for bringing this point to our attention. In the *Section Preliminary insights and perspectives on model development*, we have included a discussion on the influence of the self-stress of dislocations on local Li concentrations and diffusion heterogeneity, and suggested potential improvements for future work (Below; See Page 14, line 32-45; Page 15, line 1-4).

“Dislocation mediated Li diffusion. Bragg coherent diffraction imaging characterisation [21, 22] and chemo-mechanical phase-field dislocation modeling [64, 79, 80] indicate that discrete dislocations can result in large lattice mismatch and non-uniform stress fields near dislocations in electrode materials. These stress-strain fields around dislocations can potentially alter the overall Li diffusion behaviour and Li concentration distribution [64, 79, 80]. For example, chemo-mechanical phase-field simulations have demonstrated that there is strong Li enrichment and depletion in the tensile and compressive stress fields around the dislocation, respectively, in LiMn_2O_4 cathodes [79]. Additionally, pipe diffusion can be initiated on the tensile stress side of edge dislocations [79]. Moreover, recent advancements in strain engineering have employed dislocations to tune ionic transport properties in Li-ion conducting argyrodites $\text{Li}_6\text{PS}_5\text{Br}$ [81]. These findings suggest that inducing internal strain and dislocations in the structure of argyrodites through applying uniaxial and hydrostatic pressures can promote Li-ion disorder, and lead to higher Li-ion conductivity [81]. Although the current model in this study does not explicitly describe the effect of self-stress around dislocations on Li-ion transport behaviour in electrodes and solid electrolytes, this effect can be incorporated into the chemo-mechanical model by reformulating the diffusivity tensor as a function of plastic shear.”

Comment 5: *Why the dislocation activity is more significant in polycrystals? Since polycrystals consist of smaller primary particles, it is believed that the formation of misfit dislocations in primary particles in smaller sizes is less favorable from an energy point of view [International Journal of Mechanical Sciences, 2020, 183: 105812.]. Some discussion on the numerical prediction must be made after presenting these results.*

Response: We agree that the dimensions and geometries of cathode particles remarkably impact the formation of misfit dislocations in cathodes, as revealed by chemo-mechanical phase-field dislocation modeling and operando synchrotron X-ray diffraction experiments [1–3]. Energy-based stability analysis of misfit dislocations reveals that the minimum critical size for dislocation-free LiFePO_4 particles is around 47 nm; below this size, particles are unlikely to host a misfit dislocation at the phase boundary [2]. Synchrotron X-ray diffraction experiments indicate that large misfit strains can be effectively circumvented in electrodes comprising V_2O_5 nanospheres with diameters of 49 nm [3]. In the current study, the primary particles in polycrystalline cathodes range from 300 nm to 1 μm in diameter, while single crystal particles range from 2 μm to 12 μm . Thus, the size of the cathode particles is above the minimum critical size of dislocation-free particles [1–3]. Consequently, in this study, plastic shear in the cathodes is mainly driven by anisotropic and heterogeneous compositional strains. The particle size-dependent effect on the stability of misfit dislocations should be incorporated into the developed chemo-mechanical model, when the dimension of the electrode particles approaches the critical size [1–3]. We have included the above discussion in the *Section Role of microstructure in rate performance and defect heterogeneity* (Page 9, line 11-13; page 10, line 1-21).

Comment 6: *It is important to understand how dislocations cause degradation of the material, but recent studies show dislocations are also an excellent tool to improve the performance of the battery materials [Journal of the American Chemical Society, 2024, 146(2): 1710-1721.]. Does the author’s model also predict similar effects of dislocations for the NMC electrode materials?*

Response: We thank the reviewer for bringing this point to our attention. In the present study, the effect of dislocations on Li-ion transport was not explored. We agree that this is an interesting point to be explored in the future, specifically employing strain engineering to tune ionic transport properties in cathodes and solid electrolytes. We have added this discussion in the *Section Preliminary insights and perspectives on model development* (Below; See Page 14, line 41-45).

“Moreover, recent advancements in strain engineering have employed dislocations to tune ionic transport properties in Li-ion conducting argyrodites $\text{Li}_6\text{PS}_5\text{Br}$ [81]. This research suggests that inducing internal strain and dislocations in the structure of argyrodites through applying uniaxial and hydrostatic pressures can promote Li-ion disorder, and lead to higher Li-ion conductivity [81].”

Comment 7: *While this manuscript is based on theoretical models, the model details are very limited, e.g. the explicit formulation of chemical potential is not given in Eq. (7). There are also no details of the numerical models. Without these details, it can be difficult to follow for potential readers. I would ask the author to provide the details of the model.*

Response: We acknowledge the reviewer’s concern about the lack of model details. To fully comply, we have now expanded the manuscript to include a comprehensive description of the chemo-mechanical model in the *Section Methods* (Page 20-25), including the explicit formulation of the chemical potential in Eq. 8 (Page 21) and the numerical solution of the governing equations (Page 24-25).

Comment 8: *The author should provide the declaration for all quantities presented in the manuscript, for example, what are the definitions of the normalized and kinetic capacity loss in the numerical model?*

Response: The revised manuscript now includes a comprehensive declaration of all quantities presented (Page 6, line 2-10).

Comment 9: *What is the numerical method for solving the governing equations and the numerical model is implemented in which software?*

Response: We have now included a detailed description of the numerical methods used to solve the governing equations in the new *Section Numerical implementation* of the revised manuscript (Page 24-25). The developed chemo-mechanical model was implemented in the freeware simulation package DAMASK [4]. A large-scale parallel finite element solver

using the PETSc numerical library [5] was developed to handle the discretization and numerical solution of the coupled governing equations [6, 7]. The self-consistent solution of the coupled fields is achieved using a staggered iteration method for each time increment. This staggered iteration method allows the solution scheme of each field to be prescribed independently. The mechanical equilibrium equation is solved using an inexact Newton method with a critical point secant line search [8]. For the reaction-diffusion equation, the Newton method with a backtracking line search [9] is employed. Within each Newton iteration for both equations, a flexible GMRES linear solver [10] preconditioned with a smoothed-aggregation algebraic multigrid method [11] is employed for the linear solve.

Comment 10: *In Fig. 3, the symbol P is used for the first Piola-Kirchhoff stress in the stress equilibrium, but the symbol σ is used for Cauchy stress in the free energy which is not declared in the Methods. This is very confusing to the reviewer regarding the numerical implementation of the theoretical model.*

Response: We apologize for the confusion due to the lack of details in the model description and numerical implementation. As mentioned in our response to Comment 7, we have expanded the manuscript to include a comprehensive description of the chemo-mechanical model and the numerical solution of the governing equations in the *Section Methods* (Page 20-25).

Comment 11: *In Eq. (7), the variation should be with respect to physical quantities rather than a dimensionless quantity c ?*

Response: We appreciate the reviewer’s thorough review. Yes, indeed, the chemical potential should be the variational derivative of the total free energy with respect to the Li concentration, instead of the dimensionless Li occupancy fraction. We have revised the expression of the chemical potential in the manuscript accordingly (Page 21, Eq. 8).

Reviewer 3

Opening comment: *Authors developed chemo-mechanical model to understand the degradation of polycrystalline NMC-811 particle embedded in solid electrolyte. They utilized crystal plasticity theory to simulate the elastoplastic behavior of cathode particles; however, the solid electrolyte is purely elastic. Secondary particle size, C-rates and elastic modulus of SE are varied to understand their effect on the chemo-mechanical behavior of polycrystalline NMC-811 cathode particle. Even though the formulation has novelty of having crystal plasticity and oxygen loss dynamics, the results about using small particle size and low C-rates are useful in mitigating the capacity loss are very well reported in the literature. To improve the quality of the manuscripts, kindly consider addressing following comments. There are also some major flaws in understanding the literature which are also mentioned below.*

Response: We appreciate the reviewer for the careful reading and the constructive comments on the manuscript. Following the reviewer’s suggestions, we have now carefully

revised the manuscript at several places accordingly. As pointed out by the reviewer, we have significantly expanded existing chemo-mechanical models by incorporating crystal plasticity to study the role of dislocations in the degradation of NMC cathodes at the grain level. To the authors' knowledge, this is the first time a chemo-mechanical model has been developed to assess how microstructure and (dis)charging protocols affect the formation of dislocations and defect-induced structural degradation at the grain level in Ni-rich composite cathodes. While previous studies have reported the benefits of using small particles and low C-rates, our model offers a detailed quantitative analysis of the correlation between cycling protocols, microstructure, stress response, dislocation formation, and structural degradation. The novelty of the current work has also been highlighted by the other reviewers. For example, Reviewer 2 stated, "Some interesting topics are discussed in detail, particularly the impact of dislocations on structural degradation and capacity loss of the battery material from theoretical and numerical points of view. There is no doubt this work extended the understanding of the degradation mechanism in electrode materials at the microstructure level." Similarly, Reviewer 1 noted, "The manuscript considers scientifically interesting and practically important effects to elucidate the effects of alternative crystalline structures and surely forms the basis for a highly relevant paper."

Comment 1: *The authors mentioned in Page 2, line 12 that volume change for NMC-811 is 7.8% without any reference. In the article by Yoon et al., 2006 (<https://www.sciencedirect.com/science/article/pii/S138824810600227X>), the volume change for NMC-811 was measured to be ~3.4%. Can authors please comment on how they found 7.8% volume expansion?*

Response: We would like to clarify that the anisotropic lattice dimensional changes induced by Li intercalation are highly dependent on the chemical composition of NMC oxide cathodes. The reference suggested by the reviewer pertains to NMC111, not NMC811. The volume change for NMC111 was reported to be around 3.4% in the suggested reference. The atomic lattice dimension changes for NMC811 cathodes during (dis)charge, which were used in this study, were measured by operando synchrotron X-ray diffraction (XRD) experiments [12–14]. As reported in these references [12–14], the a and b lattice parameters increase upon lithiation by maximum 2%. The c lattice parameter exhibits a nonmonotonic behaviour, rapidly increasing at the initial stage of discharge by up to 4%, and then gradually collapsing to 1.95%. The measured volume change for NMC811 is around 7.8% [12–14]. We have added the related references on Page 2, line 12-13 in the revised manuscript.

Comment 2: *About the authors comment on Page 2, line 22, "Although the exact nature of the defect-induced bonding environment change is debated and may vary among different materials, such as Ni-rich and Li-rich layered oxides, the large lattice strain associated with crystal defects can markedly reduce the energy barriers to remove lattice oxygen and trigger Li/TM ion mixing [4, 20, 26–29]. Therefore, besides the electrochemical instability of the cathode/electrolyte interface, the continuous accumulation of crystal defects in cathode*

particles due to Li (de)intercalation over cycling acts as the primary driving force for aggravating oxygen loss and altering the structural stability.” Oxygen loss and structural change mostly happens if voltages > 4.2 (see below review article for the detail). Can authors please clarify if oxygen loss and structural change is significant enough within the operating voltage (i.e., < 4.2)? <https://onlinelibrary.wiley.com/doi/full/10.1002/aenm.202103712>

Response: We agree with the reviewer that oxygen loss and structural degradation are more prominent at higher voltages for NMC cathodes. However, as reported in the references [15–17], the onset potential of oxygen loss highly depends on the Ni content in NMC cathodes. The transition from a layered to rock-salt phase in NMC layered cathodes is an inevitable result of cationic mixing and oxygen loss [15]. Among Ni, Co, and Mn ions, Ni ions have the strongest tendency to mix with Li ions, primarily due to the similarity in ionic radius between Ni and Li ions [15, 16]. The onset potential of oxygen loss decreases with increasing Ni content, for example, at around 4.6 V for NMC111 and at 4.2 V for NMC811 [16, 17]. This indicates that Ni-rich cathodes, such as NMC811, can undergo severe structural degradation even at lower charging voltages compared to NMC111 cathodes. To provide clarity, we have included this discussion in the *Section Discussion* in the revised manuscript (Page 17, line 12-19).

Comment 3: *About the author’s statement on Page 2 and line 40. “However, the assessment of how microstructure and (dis)charging protocols affect micromechanics and the formation of crystal defects in composite cathodes remains unexplored. This gap in knowledge is primarily due to the limitations and efforts associated with the application of advanced and standardized analytical techniques in probing cathodes and their environments across various length scales [4, 5].” Can author clarify this literature gap as there are already several papers listed below understanding the chemo-mechanical behavior of NMC particles with or without solid electrolyte for different C-rates, grain sizes and interface energies?*
<https://www.sciencedirect.com/science/article/pii/S0378775321009216>
<https://www.sciencedirect.com/science/article/pii/S0378775323014106>
<https://www.sciencedirect.com/science/article/pii/S0378775324000053>
<https://www.sciencedirect.com/science/article/pii/S0022509622000540>
<https://www.sciencedirect.com/science/article/pii/S0022509621002556>

Response: We appreciate the reviewer bringing this to our attention. The revised manuscript now acknowledges the existing literature on the topic and clarifies our specific contribution. While there are several studies on the chemo-mechanical behaviour of NMC cathodes, our work specifically focuses on the detailed assessment of how microstructure and charging protocols affect the formation of crystal defects and defect-induced structural degradation at the grain level in composite cathodes. This distinguishes our work from previous studies. To address this, we have revised the text as follows (Page 2, line 41-45; Page 3, line 1):

“Chemo-mechanical simulations using cohesive zone models and phase-field damage models have advanced the understanding of the microscopic mechanical

fracture behaviour in electrode materials [32–44]. However, the assessment of how microstructure and (dis)charging protocols affect the formation of crystal defects and defect-induced structural degradation at the grain level in composite cathodes remains unexplored.”

Comment 4: *The author’s statement on Page 4 and line 2 is wrong. “As 2 characterized by electron backscatter diffraction measurements [30], the NMC811 polycrystalline secondary particles exhibit randomly distributed crystallographic orientations (Fig. 3b).” Actually, NMC-811 has radially oriented grains and in the reference provided by the authors, the same has been mentioned. The reviewer advised authors to carefully read the literature before making such statements. With this statement being wrong, the authors need to revise all the results provided in the paper.*

Response: Thank you for pointing out this oversight. The microstructure, including the morphology and texture of the NMC cathodes, can be effectively tuned by various synthesis processes [18–21]. NMC secondary particles synthesized by the co-precipitation method typically consist of many randomly oriented primary particles [18, 19]. However, modifying the surface energies through a boron doping strategy can induce the directional growth of primary particles, resulting in NMC secondary particles with radially aligned primary particles [20, 21]. In this study, an isolated Ni-rich NMC811 polycrystal particle, consisting of 200 randomly oriented primary particles, is embedded in the uniform solid electrolyte. We have revised the references and text accordingly in the manuscript to reflect this (Page 4, line 18-23).

Comment 5: *As authors only considered isotropic elastic deformation for the solid electrolyte. Can authors please comment on it as solid electrolyte also shows elasto-plastic and rate dependent behavior as reported by Papakyriakou et al., J. Power Sources (2021) and Athanasiou et al., Cell Report Phys. Science (2022)?*

Response: We thank the reviewer for bringing this to our attention. In the revised manuscript, we have included new results in Fig. 6 and in the *Section Preliminary insights and perspectives on model development* to study the effect of plastic deformation in solid electrolytes on the chemo-mechanical behaviour of composite cathodes (Below, See Page 14, line 12-31; Page 15, Fig. 6). Our findings indicate that oxide-based electrolytes exhibit minimal plastic deformation upon lithiation of composite cathodes. The impact of plastic deformation in oxide electrolytes on the formation of dislocations in cathodes and interfacial stress distribution is negligible. In contrast, plastic deformation in sulfide electrolytes, while not significantly affecting dislocation formation in NMC cathodes, effectively reduces tensile stress concentration at the interface between cathodes and solid electrolytes.

“Elasto-plastic deformation in solid electrolytes. The assessment of the chemo-mechanical behaviour of composite cathodes in this study assumes that the solid electrolyte behaves as an elastic solid, which is valid for oxide electrolytes [74, 75]. However, sulfide electrolytes are expected to undergo plastic

deformation in response to the volume change of electrodes due to their low yield strength [73-75]. For example, indentation measurements indicate that the yield strength for amorphous and crystalline sulfide electrolytes ranges from 200 MPa to 450 MPa, while the yield strength for oxide-based electrolytes is significantly higher, ranging from 2 GPa to 3 GPa [73-75]. Fig. 6c-f show the effect of the plastic deformation in oxide and sulfide electrolytes on the chemo-mechanical behaviour of composite cathodes, at a discharge rate of 1 C. Here, solid electrolytes are modeled as isotropic elasto-plastic materials using the isotropic J_2 plasticity model [48], with a yield strength of 2 GPa for oxide electrolytes, 450 MPa for crystalline sulfide electrolytes, and 200 MPa for amorphous sulfide electrolytes [73-75]. Fig. 6c shows that oxide-based electrolytes exhibit minimal plastic deformation upon lithiation of composite cathodes. The impact of plastic deformation in oxide electrolytes on the formation of dislocations in cathodes and interfacial stress distribution is negligible (Fig. 6d). In contrast, Fig. 6e, f show that while the plastic deformation in sulfide electrolytes does not obviously impact dislocation formation in NMC cathodes, it can effectively reduce the tensile stress concentration at the interface between cathodes and solid electrolytes.”

Comment 6: *On page 4, line 25, the authors mentioned that “The material domains with a plastic shear exceeding 12% in cathode particles after discharge are categorized as the oxygen-deficient phase.” Can authors please clarify how such number is decided?*

Response: The threshold value for categorizing material domains with a plastic shear exceeding 12% as the oxygen-deficient phase is based on published atomic-scale calculations [22-25] and experimental characterisation [26]. Atomic-scale calculations reveal that the formation energy of oxygen vacancies in layered oxide cathodes is significantly reduced when the applied tensile strain approaches 10% [22-25]. This threshold value is further validated by fitting the predicted distribution of oxygen-deficient phase in the secondary particle to experimental characterisation [26] (Fig. 5c in the manuscript). We have included this clarification in the revised manuscript (Page 5, line 3-10).

Comment 7: *Authors simulated three types of simulations i.e., isotropic, anisotropic Li diffusion inside the primary particle with a constant Li diffusivity, and anisotropic diffusion with a Li concentration-dependent diffusion coefficient. Can author please clarify the intension behind simulating such cases as only last one is the practical case?*

Response: We realize that the original presentation might have caused confusion, as also pointed out by Reviewer 1 in Comment 2. To address this, we have revised Fig. 3 (Page 7) to include comparisons of the chemo-mechanical behaviour in a cathode particle with isotropic and concentration-independent material properties, as well as single crystal and polycrystal particles with anisotropic and Li concentration-dependent material properties. By comparing these cases, we demonstrate the necessity and impact of using a more realistic model that includes anisotropic and concentration-dependent material properties.

Comment 8: *Can authors please clarify how they measure the single voltage profile for each case, as for anisotropic particle the voltage on each point of particle surface will be different?*

Response: The voltage of the cathode particle is calculated as the average value over the entire surface of the cathode particle. For clarity, we have added this explanation in the revised manuscript (Page 21, line 6-7).

Comment 9: *Better insights can be obtained from studying the actual reconstructed microstructure? Authors should consider using the reconstructed microstructure.*

Response: We agree with the reviewer. In the *Section Preliminary insights and perspectives on model development*, we have included a discussion on the construction of microstructure for the model from experimental data (Below; See Page 16, line 35-45; Page 17, line 1-7).

“Constructing microstructure from experimental data. Given the anisotropic electrochemical and mechanical properties of NMC cathodes, the present study, along with chemo-mechanical fracture simulations [43] and experimental characterisation studies [51, 52], demonstrate that regulating the morphology and crystallographic orientation of primary particles can effectively mitigate crystal defect formation, stress concentration, and microcracking in Ni-rich NMC cathodes. Such regulation can significantly improve the rate and cycling performance of composite cathodes. For example, modifying the surface energies through a boron doping strategy can induce the directional growth of primary particles, resulting in NMC secondary particles with radially aligned primary particles [51, 52]. This unique crystallographic texture allows diffusion channels to penetrate directly from the surface to the center, significantly improving the Li-ion diffusion coefficient. Moreover, this radial alignment can effectively alleviate the volume-change-induced stress and intergranular fracture in cathodes [43, 51, 52]. This representative three-dimensional microstructure for chemo-mechanical modeling can be generated based on multimodal microscopy characterisations [78, 85, 86]. Statistical representations of particle microstructures can be derived from X-ray nano-computed tomography data [85] and sub-particle grain representations can be derived from electron backscatter diffraction data [86].”

Comment 10: *Authors statement on Page 6, line 7 “Fig. 3b, c show that the large anisotropic volume change of Ni-rich layered cathodes driven by Li intercalation and the cathode microstructure heterogeneity result in substantial differences in dislocation activity and accumulation between primary particles” is unclear. As mentioned earlier by the reviewer that the volume change for NMC-811 is $\sim 3.5\%$. Please comment on how such a deformation can be considered as large deformation and from Fig. 3, it is difficult to anticipate that the numbers correspond to large deformation.*

Response: We have clarified the measurement of the lattice dimensional changes for NMC811 in the response to the Comment 1. The volume change for NMC811 is 7.8%, not 3.5%. Furthermore, we have added new Fig. S8 in the supplementary materials, which shows the distribution of Li concentration and grain morphologies at deformed configurations with displacement scaled by various factors. This helps to visualize the anisotropic volume changes.

Comment 11: *Rather than just showing max. principal stress as in Fig. 3(h), the reviewer advised to show the contour plots of von-Mises stresses.*

Response: While the main manuscript focuses on the distribution of the maximum principal stress to analyse crack formation, we appreciate the reviewer’s suggestion to also include von Mises stress for a more comprehensive analysis. Accordingly, we have added contour plots of the von Mises stresses in Fig. S6 and Fig. S7 in the supplementary materials.

Comment 12: *Section 4, page 7, the author’s observation that by using small secondary particles and lower C-rates, the capacity can be improved. Such results are obvious and have already been reported by numerous papers. Can authors please comment on the novelty of the results?*

Response: We believe there might be a misconception regarding the statement on the predicted capacity-rate trade-off, which can be improved by decreasing the secondary particle size. This statement is intended to demonstrate the good agreement between our simulations and experimental results, reflecting the effectiveness of the developed physics-based chemo-mechanical model. We did not intend to present this observation as a novel contribution of the manuscript. To clarify, we have added the following statement to the manuscript (Page 9, line 1-2): “The good agreement between simulations and experiments confirms the effectiveness of the developed physics-based chemo-mechanical model.”

Comment 13: *Concern on the results presented by authors for different modulus of solid electrolytes. As shown in Fig. 4(f), the stresses are of the order of 1.6 GPa, can material bear such stresses in reality? Please comment.*

Response: We agree with the reviewer that the predicted stress would generally be high enough to initiate crack formation. Since the explicit mechanical fracture is not considered in this study, we have now acknowledged this limitation in the *Section Theoretical framework and model setup* of the revised manuscript (Below; See Page 4, line 41-43). Additionally, in the *Section Preliminary insights and perspectives on model development*, we emphasized the need for future work to integrate mechanical fracture to enhance the description of the chemo-mechanical behaviour in solid-state batteries, and discussed existing models that account for these effects (Below; See Page 13, line 26-41).

“While the present study uses the maximum principal stress distribution to analyse contact mechanics problems, the current model does not explicitly account for mechanical fracture.”

“Crack formation in composite cathodes. One of the crucial aspects of composite cathodes in solid-state batteries is the stress response of their microstructure to lattice dimension changes induced by Li transport. As shown in Fig. 3, high tensile stresses build up along grain boundaries in cathodes and at the interface between cathodes and solid electrolytes upon electrochemical cycling. This high stress can lead to contact mechanics problems. Loss of interfacial contact within the cell obstructs Li transport in cathodes and charge-transfer reactions at the cathode/electrolyte interface, which subsequently results in resistance increase, capacity loss, and rate performance deterioration in batteries. While this study effectively captures the heterogeneous stress response and crystalline defect formation induced by compositional strains, it does not account for crack formation. Including mechanical fracture in the model would be beneficial to achieve a comprehensive understanding of the mechanical behaviour in solid-state batteries. Particle fracture and interfacial contact loss in electrode materials can be integrated into the chemo-mechanical models using cohesive zone models [32–35], spring analogy approaches [38, 39], continuum-damage models [40], and phase-field damage models [43, 44, 72].”

Comment 14: *The author’s statement on Page 10, line 3 “Fig. 5c consistently indicate that, on average, primary particles located near the exterior of the agglomerate are more susceptible to oxygen loss compared to those in the interior.” When the voltage goes beyond 4.2 for NMC-811, surface reconstruction happens and there is oxygen loss in the process. Such results have been reported by several experimental articles. Can the author please comment on the novelty of the results?*

Response: We believe this is a misconception regarding our original statement (Page 11, line 29-32), which reads: “both experiments [5] and predictions depicted in Fig. 5c consistently indicate that, on average, primary particles located near the exterior of the agglomerate are more susceptible to oxygen loss compared to those in the interior.” This statement is intended to demonstrate the good agreement between our predictions and experimental results.

More importantly, while the enrichment of oxygen-deficient phases on the exterior of secondary particles has been reported by experiments, our study reveals that this phenomenon is highly related to the distribution of crystal defects, i.e. dislocations, on the exterior of the secondary particles. We would like to clarify that “oxygen-deficient phases on the exterior of secondary particles” refer to the bulk structural degradation within the exterior of the secondary particle, not surface reconstruction due to side reactions between cathodes and electrolytes. We emphasize the role of crystal defects in leading to oxygen loss and bulk structural degradation. This aspect of our study—linking the distribution of crystal

defects to the susceptibility of oxygen loss and structural degradation—adds a novel insight into understanding the degradation mechanisms of NMC811 cathodes.

Comment 15: *On page 11, line 6, the authors found that particle size greater than 8 μm in diameter tend to exhibit significant capacity loss. Can authors please comment on any experimental evidence or if they performed any experimental investigation to validate such a case?*

Response: As explained in the response to the Comment 3 from Reviewer 2. We agree with the reviewers on the importance of supporting our numerical predictions with experimental data. The experimental analysis of active cathode material loss generally involves the formation of rock-salt phase and isolated cathode materials induced by intergranular fracture. Since this study does not explicitly consider crack formation, only qualitative comparisons can be made between predictions and corresponding experimental characterisation studies. In the *Section Dislocation-induced structural degradation and capacity loss* and *Section Discussion*, we have now included relevant experimental results that corroborate our predictions regarding the effect of the size of secondary particles and charging rate on capacity loss (Below; See Page 12, Fig. 5e; Page 13, line 1-15; Page 17, line 39-44; Page 18, line 1-7).

“In addition, Fig. 5e includes the effect of cathode particle size and cycling conditions on the loss of active cathode material from experimental characterisation [67–71]. It is noteworthy that the experimental analysis of active cathode material loss generally involves the formation of rock-salt phase and isolated cathode materials induced by intergranular fracture [67–71]. Since this study does not explicitly consider crack formation, only qualitative comparisons are made between predictions and experimental characterisations. Fig. 5e shows that the total loss of active cathode material increases with higher charging rate according to experimental characterisation, which aligns with the predicted results. For example, increasing the charging rate from 0.5 C to 6 C leads to an increase in the loss of active material from 8.5% to 16% [67–70]. Furthermore, experimental characterisation reveals that the capacity fading of cells using small cathode particles (average diameter of 11 μm) can be reduced from 22.1% to 16.6%, compared to those with large cathode particles (16 μm) [71]. This capacity fading mechanism is generally attributed to the loss of Li inventory, plating of Li, and loss of active material from both electrodes.”

“TEM analysis further supports these findings, suggesting that reducing the charge rate to below 0.1 C or decreasing the particle size to 1 μm can mitigate structural defects and microcracks, thereby enhancing the cycling stability of Ni-rich NMC cathodes [91]. Moreover, the simulation results presented in Fig. 4d and Fig. 7b indicate that employing single crystal cathodes can significantly reduce the formation of dislocations and mitigate oxygen-loss-related structural degradation in Ni-rich NMC cathodes. Differential electrochemical

mass spectroscopy characterisation and HRTEM characterisation also reveal that the formation of rock-salt phase in the bulk of cathodes is considerably reduced in single crystal Ni-rich cathodes, compared to polycrystalline cathodes [3, 92–96]. Therefore, the improved chemomechanical and cycling performance of single crystal NMC cathodes can be attributed to both the elimination of intergranular cracks and the reduction in defects-induced bulk structural degradation.”

Comment 16: *On page 11, line 36, “As shown in Fig. 6a,b, strategies including reducing the secondary particle diameter to below 4 μm and employing single crystal layered cathodes devoid of interparticle boundaries have proven effective in mitigating the capacity fading issue observed in conventional polycrystalline cathodes [6, 41].” It has been reported that using single crystal mitigate the fracture within the particle and can reduce capacity loss. Can authors please clarify the novelty of the results and if they have done any experimental investigation on validating that below 4 μm mitigate the capacity fading issue.*

Response: We have addressed the validation and comparison of our predictions with experimental results in the response to Comment 15.

While it has been reported that using single crystal cathodes can mitigate intergranular cracks and reduce capacity loss, our study provides novel insights into the role of dislocations in single crystal cathodes and their contribution to bulk structural degradation. Our results indicate that, in addition to the elimination of intergranular cracks, employing single crystal cathodes can significantly reduce the formation of dislocations and mitigate oxygen-loss-related structural degradation in Ni-rich NMC cathodes. Differential electrochemical mass spectroscopy characterisation and transmission electron microscopy characterisation also reveal that the formation of rock-salt phase in the bulk of cathodes is considerably reduced in single crystal Ni-rich cathodes, compared to polycrystalline cathodes [27–32]. Therefore, the improved chemo-mechanical and cycling performance of single crystal NMC cathodes can be attributed to both the elimination of intergranular cracks and the reduction in defects-induced bulk structural degradation. We have included this discussion in the *Section Discussion* of the revised manuscript (Page 17, line 39-44; Page 18, line 1-7).

References

- [1] S. Esmizadeh, H. Haftbaradaran, An energy-based stability analysis of misfit dislocations in two-phase electrode particles: A planar model and implications for LiFePO_4 , *International Journal of Mechanical Sciences* 183 (2020) 105812.
- [2] X. Zhou, C. Reimuth, B.-X. Xu, Phase-field simulation of misfit dislocations in two-phase electrode particles: Driving force calculation and stability analysis, *International Journal of Solids and Structures* 249 (2022) 111688.
- [3] Y. Luo, Y. Bai, A. Mistry, Y. Zhang, D. Zhao, S. Sarkar, J. V. Handy, S. Rezaei, A. C. Chuang, L. Carrillo, et al., Effect of crystallite geometries on electrochemical

- performance of porous intercalation electrodes by multiscale operando investigation, *Nature Materials* 21 (2) (2022) 217–227.
- [4] F. Roters, M. Diehl, P. Shanthraj, P. Eisenlohr, C. Reuber, S. L. Wong, T. Maiti, A. Ebrahimi, T. Hochrainer, H.-O. Fabritius, et al., DAMASK—The Düsseldorf Advanced Material Simulation Kit for modeling multi-physics crystal plasticity, thermal, and damage phenomena from the single crystal up to the component scale, *Computational Materials Science* 158 (2019) 420–478.
- [5] S. Balay, S. Abhyankar, M. F. Adams, S. Benson, J. Brown, P. Brune, K. Buschelman, E. M. Constantinescu, L. Dalcin, A. Dener, V. Eijkhout, J. Faibussowitsch, W. D. Gropp, V. Hapla, T. Isaac, P. Jolivet, D. Karpeev, D. Kaushik, M. G. Knepley, F. Kong, S. Kruger, D. A. May, L. C. McInnes, R. T. Mills, L. Mitchell, T. Munson, J. E. Roman, K. Rupp, P. Sanan, J. Sarich, B. F. Smith, S. Zampini, H. Zhang, H. Zhang, J. Zhang, [PETSc Web page, https://petsc.org/](https://petsc.org/) (2024).
URL <https://petsc.org/>
- [6] P. Shanthraj, C. Liu, A. Akbarian, B. Svendsen, D. Raabe, Multi-component chemo-mechanics based on transport relations for the chemical potential, *Computer Methods in Applied Mechanics and Engineering* 365 (2020) 113029.
- [7] C. Liu, P. Shanthraj, M. Diehl, F. Roters, S. Dong, J. Dong, W. Ding, D. Raabe, An integrated crystal plasticity–phase field model for spatially resolved twin nucleation, propagation, and growth in hexagonal materials, *International Journal of Plasticity* 106 (2018) 203–227.
- [8] S. C. Eisenstat, H. F. Walker, Choosing the forcing terms in an inexact Newton method, *SIAM Journal on Scientific Computing* 17 (1) (1996) 16–32.
- [9] J. E. Dennis Jr, R. B. Schnabel, *Numerical methods for unconstrained optimization and nonlinear equations*, SIAM, 1996.
- [10] Y. Saad, A flexible inner-outer preconditioned GMRES algorithm, *SIAM Journal on Scientific Computing* 14 (2) (1993) 461–469.
- [11] P. Vanek, J. Mandel, M. Brezina, Algebraic multigrid by smoothed aggregation for second and fourth order elliptic problems, *Computing* 56 (3) (1996) 179–196.
- [12] K. Märker, P. J. Reeves, C. Xu, K. J. Griffith, C. P. Grey, Evolution of structure and lithium dynamics in $\text{LiNi}_{0.8}\text{Mn}_{0.1}\text{Co}_{0.1}\text{O}_2$ (NMC811) cathodes during electrochemical cycling, *Chemistry of Materials* 31 (7) (2019) 2545–2554.
- [13] C. Xu, K. Märker, J. Lee, A. Mahadevegowda, P. J. Reeves, S. J. Day, M. F. Groh, S. P. Emge, C. Ducati, B. Layla Mehdi, et al., Bulk fatigue induced by surface reconstruction in layered Ni-rich cathodes for Li-ion batteries, *Nature Materials* 20 (1) (2021) 84–92.

- [14] C. Xu, P. J. Reeves, Q. Jacquet, C. P. Grey, Phase behavior during electrochemical cycling of Ni-rich cathode materials for Li-ion batteries, *Advanced Energy Materials* 11 (7) (2021) 2003404.
- [15] S. S. Zhang, Problems and their origins of Ni-rich layered oxide cathode materials, *Energy Storage Materials* 24 (2020) 247–254.
- [16] S. S. Zhang, Understanding of performance degradation of $\text{LiNi}_{0.80}\text{Co}_{0.10}\text{Mn}_{0.10}\text{O}_2$ cathode material operating at high potentials, *Journal of energy chemistry* 41 (2020) 135–141.
- [17] R. Jung, M. Metzger, F. Maglia, C. Stinner, H. A. Gasteiger, Oxygen release and its effect on the cycling stability of $\text{LiNi}_x\text{Mn}_y\text{Co}_z\text{O}_2$ (NMC) cathode materials for Li-ion batteries, *Journal of The Electrochemical Society* 164 (7) (2017) A1361.
- [18] L. Wang, B. Zhu, D. Xiao, X. Zhang, B. Wang, H. Li, T. Wu, S. Liu, H. Yu, Grain Morphology and Microstructure Control in High-Stable Ni-Rich Layered Oxide Cathodes, *Advanced Functional Materials* 33 (31) (2023) 2212849.
- [19] A. Quinn, H. Moutinho, F. Usseglio-Viretta, A. Verma, K. Smith, M. Keyser, D. P. Finegan, Electron backscatter diffraction for investigating lithium-ion electrode particle architectures, *Cell Reports Physical Science* 1 (8) (2020).
- [20] X. Xu, H. Huo, J. Jian, L. Wang, H. Zhu, S. Xu, X. He, G. Yin, C. Du, X. Sun, Radially oriented single-crystal primary nanosheets enable ultrahigh rate and cycling properties of $\text{LiNi}_{0.8}\text{Co}_{0.1}\text{Mn}_{0.1}\text{O}_2$ cathode material for lithium-ion batteries, *Advanced Energy Materials* 9 (15) (2019) 1803963.
- [21] K.-J. Park, H.-G. Jung, L.-Y. Kuo, P. Kaghazchi, C. S. Yoon, Y.-K. Sun, Improved cycling stability of $\text{Li}[\text{Ni}_{0.90}\text{Co}_{0.05}\text{Mn}_{0.05}]\text{O}_2$ through microstructure modification by boron doping for Li-ion batteries, *Advanced energy materials* 8 (25) (2018) 1801202.
- [22] A. Singer, M. Zhang, S. Hy, D. Cela, C. Fang, T. Wynn, B. Qiu, Y. Xia, Z. Liu, A. Ulvestad, et al., Nucleation of dislocations and their dynamics in layered oxide cathode materials during battery charging, *Nature Energy* 3 (8) (2018) 641–647.
- [23] T. Liu, J. Liu, L. Li, L. Yu, J. Diao, T. Zhou, S. Li, A. Dai, W. Zhao, S. Xu, et al., Origin of structural degradation in Li-rich layered oxide cathode, *Nature* 606 (7913) (2022) 305–312.
- [24] J.-H. Song, S. Yu, B. Kim, D. Eum, J. Cho, H.-Y. Jang, S.-O. Park, J. Yoo, Y. Ko, K. Lee, et al., Slab gliding, a hidden factor that induces irreversibility and redox asymmetry of lithium-rich layered oxide cathodes, *Nature Communications* 14 (1) (2023) 4149.

- [25] Q. Li, Z. Yao, E. Lee, Y. Xu, M. M. Thackeray, C. Wolverton, V. P. Dravid, J. Wu, Dynamic imaging of crystalline defects in lithium-manganese oxide electrodes during electrochemical activation to high voltage, *Nature Communications* 10 (1) (2019) 1692.
- [26] P. M. Csernica, S. S. Kalirai, W. E. Gent, K. Lim, Y.-S. Yu, Y. Liu, S.-J. Ahn, E. Kaeli, X. Xu, K. H. Stone, et al., Persistent and partially mobile oxygen vacancies in Li-rich layered oxides, *Nature Energy* 6 (6) (2021) 642–652.
- [27] X. Liu, G.-L. Xu, V. S. C. Kolluru, C. Zhao, Q. Li, X. Zhou, Y. Liu, L. Yin, Z. Zhuo, A. Daali, et al., Origin and regulation of oxygen redox instability in high-voltage battery cathodes, *Nature Energy* 7 (9) (2022) 808–817.
- [28] C. Wang, R. Zhang, C. Siu, M. Ge, K. Kisslinger, Y. Shin, H. L. Xin, Chemomechanically stable ultrahigh-Ni single-crystalline cathodes with improved oxygen retention and delayed phase degradations, *Nano Letters* 21 (22) (2021) 9797–9804.
- [29] M. Yi, J. Li, X. Fan, M. Bai, Z. Zhang, B. Hong, Z. Zhang, G. Hu, H. Jiang, Y. Lai, Single crystal Ni-rich layered cathodes enabling superior performance in all-solid-state batteries with PEO-based solid electrolytes, *Journal of Materials Chemistry A* 9 (31) (2021) 16787–16797.
- [30] E. Hu, Y. Lyu, H. L. Xin, J. Liu, L. Han, S.-M. Bak, J. Bai, X. Yu, H. Li, X.-Q. Yang, Explore the effects of microstructural defects on voltage fade of Li-and Mn-rich cathodes, *Nano Letters* 16 (10) (2016) 5999–6007.
- [31] T. Bartsch, F. Strauss, T. Hatsukade, A. Schiele, A.-Y. Kim, P. Hartmann, J. Janek, T. Brezesinski, Gas evolution in all-solid-state battery cells, *ACS Energy Letters* 3 (10) (2018) 2539–2543.
- [32] J. Hu, L. Li, Y. Bi, J. Tao, J. Lochala, D. Liu, B. Wu, X. Cao, S. Chae, C. Wang, et al., Locking oxygen in lattice: A quantifiable comparison of gas generation in polycrystalline and single crystal Ni-rich cathodes, *Energy Storage Materials* 47 (2022) 195–202.

REVIEWERS' COMMENTS

Reviewer #1 (Remarks to the Author):

The revisions are satisfactory, and I recommend the paper for publication.

Reviewer #2 (Remarks to the Author):

The authors carefully responded to my comments and revised the manuscript. The manuscript now is much better well written than the previous version. However, in response to comment 3 in the revised manuscript, the reviewer feels the experiment results from the existing papers still provide limited support for validating the numerical results because the experiment results contain the influence of all kinds of defects. The author stated in the revised manuscript, “Therefore, the improved chemomechanical and cycling performance of single crystal NMC cathodes can be attributed to both the elimination of intergranular cracks and the reduction in defects-induced bulk structural degradation.”. This is very true and an obvious conclusion. But it is still unclear to the reviewer, under which circumstances, the studied structural defects (dislocations and oxygen losses) can become a dominant reason to influence the performance of the cathode materials, or how to distinguish the contributions to capacity losses between dislocations and cracks. For example, if the particle size becomes small, the crack formation becomes less possible and dislocations still nucleate, does the capacity loss induced by dislocations and oxygen losses can be matched better with experiment results? This part of the discussion should be emphasized as the novelty of the numerical results in addition to the novelty of the theoretical model.

Reviewer #3 (Remarks to the Author):

The authors have significantly revised the manuscript by adding the missing literature, formulation, and assumptions. They have addressed most of the comments made by the reviewer.

Responses to reviewers' comments on "Role of grain-level chemo-mechanics in composite cathode degradation of solid-state lithium batteries"

Chuanlai Liu, Franz Roters, Dierk Raabe

We appreciate the editor and reviewers for their time and input in reviewing our manuscript. The comments suggested by the reviewers are addressed below, and changes are highlighted in orange in the revised manuscript.

Reviewer 1

Comment: *The revisions are satisfactory, and I recommend the paper for publication.*

Response: We cordially thank the reviewer for the appreciation of our revision work.

Reviewer 2

Comment: *The authors carefully responded to my comments and revised the manuscript. The manuscript now is much better well written than the previous version. However, in response to comment 3 in the revised manuscript, the reviewer feels the experiment results from the existing papers still provide limited support for validating the numerical results because the experiment results contain the influence of all kinds of defects. The author stated in the revised manuscript, "Therefore, the improved chemomechanical and cycling performance of single crystal NMC cathodes can be attributed to both the elimination of intergranular cracks and the reduction in defects-induced bulk structural degradation.". This is very true and an obvious conclusion. But it is still unclear to the reviewer, under which circumstances, the studied structural defects (dislocations and oxygen losses) can become a dominant reason to influence the performance of the cathode materials, or how to distinguish the contributions to capacity losses between dislocations and cracks. For example, if the particle size becomes small, the crack formation becomes less possible and dislocations still nucleate, does the capacity loss induced by dislocations and oxygen losses can be matched better with experiment results? This part of the discussion should be emphasized as the novelty of the numerical results in addition to the novelty of the theoretical model.*

Response: We appreciate the reviewer for carefully reviewing this manuscript and affirming the improvement of the revised manuscript. We agree that the experimental analysis of active cathode material loss in polycrystal cathodes contains the formation of rock-salt

phase and isolated cathode materials induced by intergranular fracture. To address the reviewer’s concern about the impact of crystal defects on performance, we emphasize that the role of dislocation formation in the structural degradation of NMC cathodes can be more clearly illustrated in single crystal cathodes, due to their high resistance to microcrack formation. Our simulation results in Fig. 4d in the revised manuscript show that a high discharge rate of 5 C still results in a spatially heterogeneous distribution of Li-ions inside the single crystal cathode particle, which induces a large strain gradient and triggers the accumulation of dislocations at the edges of single crystal particles, despite the absence of intergrain boundaries in these cathodes. Multiscale spatial resolution diffraction and imaging experiments [1] also reveal that these structural defects cannot be eliminated simply with Li-ion deintercalation or reinsertion in NMC single crystal cathodes. The accumulation of these unrecoverable crystal defects through repeated cycling at high charge rates exacerbates irreversible phase transformation, ultimately leading to the electrochemical decay of single crystal cathodes. In full compliance with the reviewer’s pertinent suggestion we have now added these discussion items in the *Section Discussion* (Below; See Page 14, line 37-44; Page 15, line 1-7).

“It is noteworthy that, as shown in Fig. 4d, a high discharge rate of 5 C still results in a spatially heterogeneous distribution of Li-ions inside the single crystal cathode particle, which induces a large strain gradient and triggers the accumulation of dislocations at the edges of single crystal particles, despite the absence of intergrain boundaries in these cathodes. As characterised by the multiscale spatial resolution diffraction and imaging experiments [97], these structural defects cannot be eliminated simply with Li-ion deintercalation or reinsertion in NMC single crystal cathodes. The accumulation of these unrecoverable crystal defects through repeated cycling at high charge rates exacerbates irreversible phase transformation, ultimately leading to the electrochemical decay of single crystal cathodes. To mitigate dislocation formation, it is crucial to enhance Li-ion diffusion kinetics and suppress heterogeneous electrochemical reactions. This can be accomplished by reducing the size of such single crystal cathodes, regulating the crystal facets to reduce Li-ion diffusion pathways, modifying the crystal structure to minimize Li or Ni antisite disordering, and improving electronic conductivity [97].”

Reviewer 3

Comment: *The authors have significantly revised the manuscript by adding the missing literature, formulation, and assumptions. They have addressed most of the comments made by the reviewer.*

Response: We are grateful for the appreciation of the revised manuscript.

References

- [1] W. Huang, T. Liu, L. Yu, J. Wang, T. Zhou, J. Liu, T. Li, R. Amine, X. Xiao, M. Ge, et al., Unrecoverable lattice rotation governs structural degradation of single-crystalline cathodes, *Science* 384 (6698) (2024) 912–919.